# Xbp1s-FoxO1 axis governs lipid accumulation and contractile performance in heart failure with preserved ejection fraction

Gabriele G. Schiattarella [1,2,3,4,5], Francisco Altamirano [1], Soo Young Kim[1], Dan Tong[1], Anwarul Ferdous [1], Hande Piristine [1], Subhajit Dasgupta[1], Xuliang Wang[1], Kristin M. French[1], Elisa Villalobos[1], Stephen B. Spurgin [6], Maayan Waldman[1], Nan Jiang[1], Herman I. May [1], Theodore M. Hill[1], Yuxuan Luo[1], Heesoo Yoo[1], Vlad G. Zaha [1,7,8,9], Sergio Lavandero [1,10], Thomas G. Gillette [1] & Joseph A. Hill [1,11✉]

Heart failure with preserved ejection fraction (HFpEF) is now the dominant form of heart failure and one for which no efficacious therapies exist. Obesity and lipid mishandling greatly contribute to HFpEF. However, molecular mechanism(s) governing metabolic alterations and perturbations in lipid homeostasis in HFpEF are largely unknown. Here, we report that cardiomyocyte steatosis in HFpEF is coupled with increases in the activity of the transcription factor FoxO1 (Forkhead box protein O1). FoxO1 depletion, as well as over-expression of the Xbp1s (spliced form of the X-box-binding protein 1) arm of the UPR (unfolded protein response) in cardiomyocytes each ameliorates the HFpEF phenotype in mice and reduces myocardial lipid accumulation. Mechanistically, forced expression of Xbp1s in cardiomyocytes triggers ubiquitination and proteasomal degradation of FoxO1 which occurs, in large part, through activation of the E3 ubiquitin ligase STUB1 (STIP1 homology and U-box-containing protein 1) a novel and direct transcriptional target of Xbp1s. Our findings uncover the Xbp1s-FoxO1 axis as a pivotal mechanism in the pathogenesis of cardiometabolic HFpEF and unveil previously unrecognized mechanisms whereby the UPR governs metabolic alterations in cardiomyocytes.

[1] Department of Internal Medicine (Cardiology), University of Texas Southwestern Medical Center, Dallas, TX, USA. [2] Department of Advanced Biomedical Sciences, Federico II University, Naples, Italy. [3] Center for Cardiovascular Research (CCR), Department of Cardiology, Charité - Universitätsmedizin Berlin, Berlin, Germany. [4] DZHK (German Centre for Cardiovascular Research), Partner Site Berlin, Berlin, Germany. [5] Translational Approaches in Heart Failure and Cardiometabolic Disease, Max Delbrück Center for Molecular Medicine in the Helmholtz Association (MDC), Berlin, Germany. [6] Department of Pediatrics, University of Texas Southwestern Medical Center, Dallas, TX, USA. [7] Advanced Imaging Research Center, University of Texas Southwestern Medical Center, Dallas, TX, USA. [8] Harold C. Simmons Comprehensive Cancer, University of Texas Southwestern Medical Center, Dallas, TX, USA. [9] Parkland Health & Hospital System, Dallas, TX, USA. [10] Advanced Center for Chronic Diseases (ACCDiS), Faculty of Chemical & Pharmaceutical Sciences & Faculty of Medicine, Universidad de Chile, Santiago, Chile. [11] Department of Molecular Biology, University of Texas Southwestern Medical Center, Dallas, TX, USA. ✉email: joseph.hill@utsouthwestern.edu

Heart failure with preserved ejection fraction (HFpEF) is now the most prevalent form of HF worldwide[1–3]. In striking contrast with the other major form of HF—HF with reduced ejection fraction, HFrEF—the incidence and prevalence of HFpEF have increased steadily over the last decade. As of today, more than two-thirds of patients with HF over 65 years of age suffer from the clinical syndrome of HFpEF[3]. Despite the enormity of the problem, efficacious treatment options for these millions of individuals are lacking[4].

Limited understanding of pathophysiological mechanisms underlying HFpEF is a prominent reason for the failure of clinical trials thus far. HFpEF is a heterogeneous syndrome and not all patients with HFpEF harbor the same predisposing conditions[4]. However, the vast majority of them present with obesity and metabolic syndrome[5–8]. Obesity increases the risk of HFpEF over HFrEF and is associated with worsening of functional parameters in HFpEF[6]. Indeed, elements of metabolic alteration have been identified as crucial pathophysiological drivers of HFpEF in both preclinical models and clinical studies[5,9,10]. Despite the well-recognized association between obesity and HFpEF, mechanistic elements governing the excess lipid accumulation in cardiomyocytes—i.e. cardiac lipid overload—are poorly understood.

Disruption of endoplasmic reticulum (ER) homeostasis—also known as ER stress—is a common feature in disease-stressed cardiomyocytes[11]. ER stress triggers the unfolded protein response (UPR), a multidimensional, adaptive signaling pathway capable of mitigating stress under many conditions. Despite the name, stresses other than unfolded proteins can activate the UPR. Current evidence points to a clear association between the UPR and hallmarks of metabolic syndrome: obesity and dyslipidemia[12]. Indeed, either prolonged dysregulation of lipid synthesis or failure of UPR activation to resolve insults can disrupt the homeostatic ER environment and promote disease development[13,14].

In mammals, the UPR involves three ER transmembrane protein sensors: IRE1α (inositol-requiring kinase 1α), PERK (PKR-like endoplasmic reticulum kinase), and ATF6 (activating transcription factor 6). Differential roles of the individual UPR transducers have been proposed. The IRE1α pathway is the most highly conserved branch of the UPR. Upon ER stress, IRE1α initiates the unconventional splicing of mRNA encoding Xbp1 (X-box–binding protein 1) to generate spliced Xbp1 (Xbp1s), a powerful transcription factor involved in many cellular functions essential to stress responsiveness. We described a cardioprotective role of Xbp1s in HFpEF, providing evidence of IRE1α/Xbp1s dysregulation in cardiomyocyte function in both experimental and clinical HFpEF[9]. Across the spectrum of cardiovascular diseases, suppression of the IRE1α/Xbp1s arm of the UPR is an alteration unique to HFpEF. Indeed, restoring Xbp1s in HFpEF cardiomyocytes substantially ameliorates the syndrome in mice[9,10]. Looking forward, how disruption of the UPR contributes to lipotoxic alterations in cardiomyocytes and cardiometabolic disease is poorly characterized, and mechanisms whereby Xbp1s ameliorates the HFpEF phenotype remain elusive. Hence, we set out to define and manipulate mechanisms downstream of Xbp1s in cardiometabolic HFpEF and decipher its cardioprotective actions.

Here, we show that myocardial lipid accumulation and cardiac metabolic alterations in HFpEF are coupled with increases in the abundance and activity of FoxO1 (Forkhead box protein O1), a conserved transcription factor involved in cell metabolism. FoxO1 governs lipid accumulation in cardiomyocytes, and overexpression of Xbp1s triggers FoxO1 ubiquitination and proteasomal degradation. Using gain- and loss-of-function approaches we demonstrate that genetic deletion of FoxO1, as well as Xbp1s overexpression, in cardiomyocytes each ameliorates the HFpEF

phenotype and reduces cardiac steatosis. Finally, we identify the E3 ubiquitin ligase STUB1 (STIP1 homology and U-Box-containing protein 1) as a novel and direct transcriptional target of Xbp1s implicated in Xbp1s-dependent degradation of FoxO1 in cardiomyocytes. These findings link UPR alterations with metabolic dysregulation in HFpEF, unveiling the Xbp1s-FoxO1 axis as a critical mechanism in the pathogenesis of lipid abnormalities in cardiomyocytes.

## Results

**Xbp1s reduces myocardial lipid accumulation in HFpEF**. We have shown previously that a murine model of HFpEF displays features of cardiometabolic disease[9,10]. As expected, cardiac tissue samples from HFpEF mice manifest a significant increase in Oil Red O staining for neutral lipids (Supplementary Fig. 1a) as well as activation of a lipogenic gene program (Supplementary Fig. 1b) in the absence of significant changes in the expression of genes involved in lipid oxidation (Supplementary Fig. 1c). Importantly, similar accumulation of neutral lipids and increased transcript abundance of lipogenic genes were detected in cardiomyocytes (adult murine ventricular myocytes—AMVMs) isolated from HFpEF hearts (Supplementary Figure 1d, e).

In murine and human HFpEF, downregulation of Xbp1s in cardiomyocytes correlates with the development of the syndrome, and cardiomyocyte-specific overexpression of Xbp1s is sufficient to ameliorate the HFpEF phenotype[9] (Supplementary Fig. 2a–e and Supplementary Table 1). In light of this, we next set out to test for a causal role of Xbp1s-associated protective effects. Qualitative staining for neutral lipids as well as measurement of triglyceride content revealed significant reduction of lipid accumulation in the hearts of cardiomyocyte-specific Xbp1s-overexpressing transgenic mice (XBP1s TG) despite exposure to HFpEF-inducing conditions (Fig. 1a, b). Xbp1s overexpression was also associated with suppression of lipid storage/transport gene expression (Supplementary Fig. 2f). Importantly, Xbp1s overexpression in XBP1s TG hearts, as well as in neonatal rat ventricular myocytes (NRVMs) using a recombinant adenovirus driving expression of Xbp1s (AdXbp1s; Supplementary Fig. 2g, h) did not alter basal mitochondrial lipid oxidation (Supplementary Fig. 2i, j), suggesting that reduced lipid levels in Xbp1s-overexpressing cardiomyocytes are likely not the result of increased lipid utilization. Collectively, these results suggest that Xbp1s regulates cardiomyocyte lipid content in HFpEF.

**Xbp1s promotes FoxO1 protein degradation in cardiomyocytes**. Prior studies in the liver have revealed a role for Xbp1s in regulating metabolism by modulating FoxO1 activity[15]. In addition, we and others have described previously a critical role of FoxO1 in regulating lipid homeostasis[16,17]. Based on this, we hypothesized an interplay between Xbp1s and FoxO1 transcription factors occurs in the heart. Notably, in the hearts of Xbp1s TG mice, we detected a significant reduction in FoxO1 protein (Fig. 1c, d), whereas FoxO1 mRNA levels under both control (CTR) and HFpEF conditions remained unchanged (Supplementary Fig. 2k). These data suggest Xbp1s-dependent post-transcriptional regulation of steady-state FoxO1 protein levels.

FoxO1 protein stability is known to be regulated by proteasomal degradation[18]. To test for this in cardiomyocytes, NRVMs were infected with AdXbp1s or adenovirus encoding green fluorescent protein (AdGFP) in the presence or absence of the proteasome inhibitor MG132. In NRVMs, Xbp1s overexpression increased FoxO1 polyubiquitination (Supplementary Fig. 3a) and reduced FoxO1 protein levels (Fig. 1e, f) and nuclear localization (Fig. 1g, h) without affecting its transcript levels (Supplementary Fig. 2l). Xbp1s-induced degradation of FoxO1

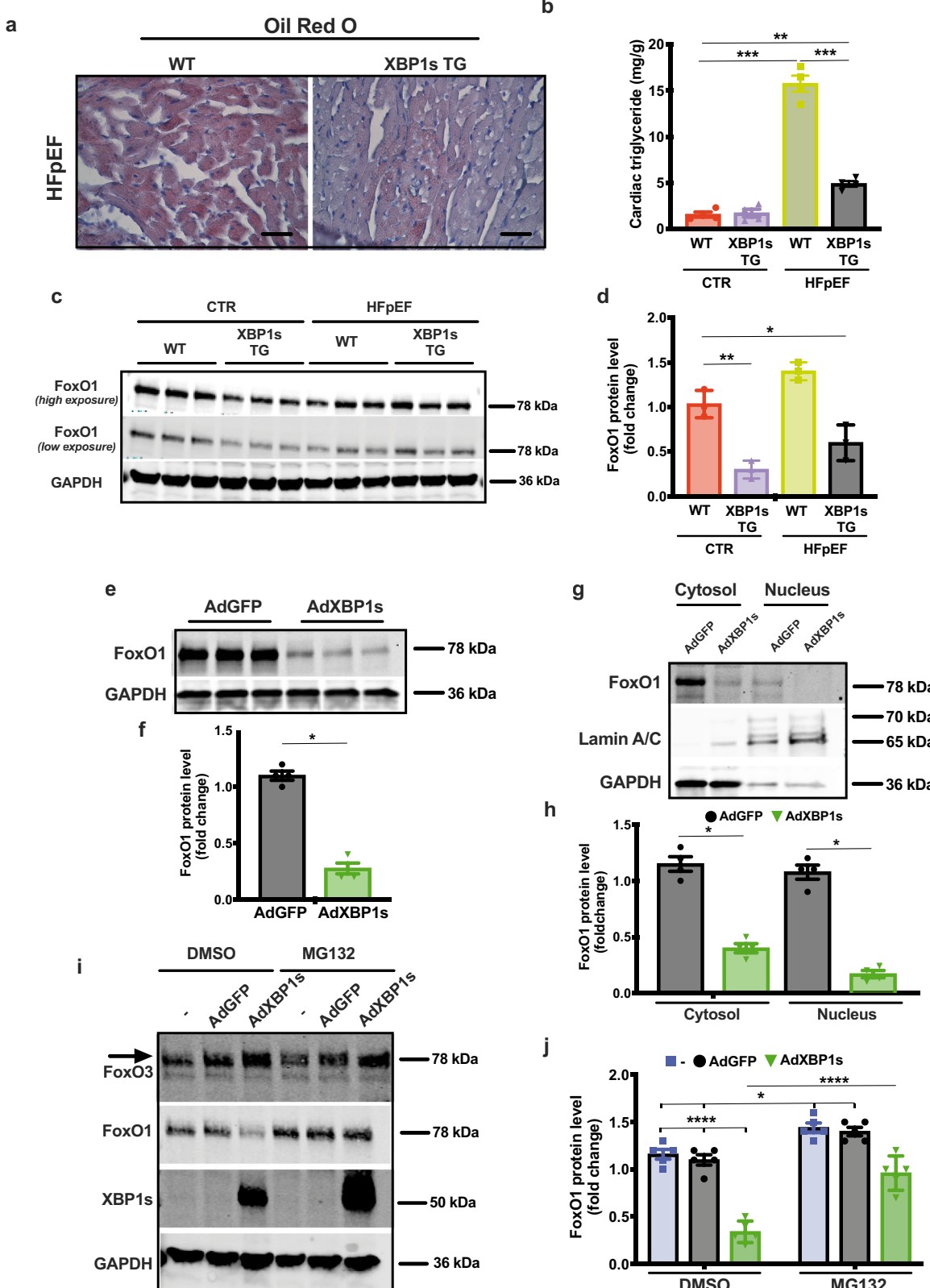

protein was partially rescued by MG132 proteasome inhibition (Fig. 1i). Importantly, protein levels of the other major FoxO family member in cardiomyocytes, FoxO3, were not affected by XBP1s overexpression (Fig. 1i, j). In aggregate, these findings establish Xbp1s as a regulator of FoxO1 protein stability in a proteasome-dependent manner.

**FoxO1 promotes lipid accumulation in HFpEF cardiomyocytes.** Our preclinical two-hit model of HFpEF stems from the combination of metabolic alterations (exposure to high-fat diet; HFD) and endothelial dysfunction-driven hypertension (modeled using the inhibitor of constitutive nitric oxide synthases N[w]-nitro-l-arginine methyl ester; L-NAME) to recapitulate the

**Fig. 1 Cardiomyocyte-specific overexpression of Xbp1s mitigates cardiac steatosis and induces FoxO1 protein degradation in HFpEF. a** Representative images of Oil Red O staining of left ventricular (LV) sections from wild-type (WT) and cardiomyocyte-restricted Xbp1s overexpressing (Xbp1s TG) HFpEF mice. Scale bars = 50 μm. Images are representative of four hearts/group. **b** Cardiac triglyceride content in myocardial tissue from WT and Xbp1s TG control (CTR) and HFpEF hearts ($n = 4$/group). **c** Representative immunoblot images of FoxO1 and GAPDH proteins from LV of WT and Xbp1s TG mice under CTR and HFpEF conditions. Images are representative of three hearts/group. **d** Densitometric analysis of FoxO1 protein band. ($n = 3$/group). **e** Representative immunoblot images of FoxO1 and GAPDH proteins from neonatal rat ventricular myocytes (NRVMs) transduced with adenovirus expressing green fluorescent protein (GFP; AdGFP) or Xbp1s (AdXbp1s). Images are representative of four independent experiments. **f** Densitometric analysis of FoxO1 protein band in the different experimental groups. ($n = 4$ biologically independent experiments). **g** Representative immunoblot images of FoxO1, Lamin A/C, and GAPDH proteins in cytosolic and nuclear extracts from NRVMs transduced with AdGFP or AdXbp1s. Images are representative of four independent experiments. **h** Densitometric analysis of cytosolic and nuclear FoxO1 protein band in the different experimental groups ($n = 4$ biologically independent experiments). **i** Representative immunoblot images of FoxO3, FoxO1, Xbp1s, and GAPDH proteins from NRVMs transduced with AdGFP, AdXbp1s, or not transduced (−) in presence or absence of MG132. Arrow indicates FoxO3-specific band. Images are representative of five independent experiments. **j** Densitometric analysis of FoxO1 protein band in the different experimental groups ($n = 5$ biologically independent experiments). Results are presented as mean ± S.E.M. In **b**, **d**, and **j**, *$p < 0.05$, **$p < 0.005$, ***$p < 0.0005$, ****$p < 0.0001$, two-way ANOVA plus Sidak's multiple comparisons test. In **f** and **h**, *$p < 0.05$, unpaired, two-tailed Kolmogorov–Smirnov test.

numerous features of clinical HFpEF[9]. Indeed, single hits, either HFD or L-NAME alone, do not elicit HFpEF in mice[9]. To evaluate FoxO1 activation in the setting of these different stresses, we assessed its nuclear localization. Nuclear fractionation of left ventricular (LV) samples revealed no increase in nuclear FoxO1 in animals treated with L-NAME alone (Supplementary Fig. 4a). Furthermore, short-term HFD was sufficient to increase nuclear FoxO1 levels (Supplementary Fig. 4a). HFpEF hearts manifested a threefold increase in FoxO1 nuclear localization when compared with regular diet CTR mice and twofold over HFD alone-treated mice (Fig. 2a, b and Supplementary Fig. 4a). A similar increase in nuclear localization of FoxO1 was observed in AMVMs isolated from HFpEF mice, confirming that these changes are cardiomyocyte-specific (Fig. 2c, d). We also observed an increase in the FoxO1-dependent transcriptional program in both HFpEF LV and AMVM samples (Fig. 2e, f). These data indicate that FoxO1 activity is increased in cardiomyocytes in the setting of HFpEF.

To determine the specific role of FoxO1 in promoting lipid accumulation in cardiomyocytes, we treated NRVMs with bovine serum albumin (BSA)/oleate mono-unsaturated fatty acid (FA) complex. Twenty-four and 48 hours of treatment were sufficient to induce lipid accumulation (Supplementary Fig. 4b) without significantly impacting cardiomyocyte viability (Supplementary Fig. 4c). In parallel, FoxO1 transcript levels were increased in NRVMs as early as six hours after BSA/Oleate treatment (Supplementary Fig. 4d). To test for a causal link between increased FoxO1 levels and lipid accumulation in cardiomyocytes, we employed both FoxO1 gain- and loss-of-function approaches. As expected, infection of NRVMs with recombinant adenovirus driving expression of a constitutively active FoxO1 mutant (AdcaFoxO1) resulted in a robust increase in FoxO1 transcript and its gene-dependent transcriptional program, whereas control infection with AdGFP had no effect (Supplementary Fig. 4e, f). AdcaFoxO1 overexpression significantly increased lipid accumulation in BSA/Oleate-treated NRVMs (Supplementary Fig. 4g). In addition, FoxO1 knockdown using FoxO1-specific small interfering RNA (siRNA; siFoxO1) (Supplementary Fig. 4h) resulted in less BSA/Oleate-induced lipid accumulation in NRVMs (Supplementary Fig. 4i). Collectively, these findings reveal previously unrecognized activation of FoxO1 in murine HFpEF and establish its role in promoting lipid accumulation in cardiomyocytes.

**Cardiomyocyte-specific FoxO1 deletion ameliorates the HFpEF phenotype and diminishes cardiac steatosis.** Based on the association between FoxO1 downregulation and the cardiometabolic protective effects observed in Xbp1s TG mice, we hypothesized that FoxO1 activation in cardiomyocytes exerts a detrimental

action in HFpEF. To test this, we studied tamoxifen-inducible, cardiomyocyte-specific FoxO1 knockout mice (cKO). FoxO1-cKO mice and littermate controls (F/F) were exposed to HFpEF or control diet for seven weeks (Fig. 3a). Both systolic and diastolic LV performance were evaluated by non-invasive echocardiography. Systolic function measured as %LV ejection fraction (LVEF%) was normal in all groups (Fig. 3b and Supplementary Table 1). As expected, F/F mice exposed to HFpEF-inducing conditions manifested increased early (E) wave to atrial (A) wave ratio (E/A) on mitral pulse Doppler as well as increased ratios of E wave to E' wave on mitral tissue Doppler (E/E'), both indicative of severe diastolic dysfunction (Fig. 3c–e and Supplementary Table 1). Under HFpEF conditions, cKO mice exhibited significantly improved diastolic function (Fig. 3c–e and Supplementary Table 1) as well as improved exercise capacity (Fig. 3f) and reduced pulmonary congestion (Fig. 3g), in the absence of significant changes in cardiac hypertrophy (Supplementary Fig. 5a, b). Importantly, cardiomyocyte-specific deletion of FoxO1 in HFpEF was associated with a reduction in myocardial neutral lipid staining (Fig. 3h), triglyceride content (Fig. 3i), and normalization of lipid gene expression (Fig. 3j). Examination of mitochondrial substrate usage revealed a decrease in cardiac pyruvate oxidation in HFpEF (Supplementary Fig. 6a) coupled with an increase in protein levels of PDK4, a negative regulator of pyruvate dehydrogenase (Supplementary Fig. 6b, c). Specificity of the PDK4 protein band was confirmed using cardiac lysate from cardiomyocyte-specific PDK4 transgenic mice (PDK4 TG)[19]. This effect was independent of FoxO1. In addition, we observed a reduction of cardiac fatty-acid oxidation in HFpEF coupled with a trend toward an increase in lipid oxidation in FoxO1-cKO mice (Supplementary Fig. 6d). In aggregate, these results point to global amelioration of the HFpEF phenotype in cKO mice, supporting a role for activation of FoxO1 in cardiomyocyte lipid accumulation and development of HFpEF.

**The E3 ubiquitin ligase STUB1 is a direct transcriptional target of Xbp1s.** Based on our findings unveiling the Xbp1s-FoxO1 axis as a critical element of HFpEF pathogenesis, we next set out to delineate mechanisms of Xbp1s-dependent FoxO1 degradation in cardiomyocytes. Ubiquitin-dependent degradation of FoxO1 can be mediated by multiple E3 ubiquitin ligases[18]. Surveying the gene promoter regions of the ubiquitin ligases known to be involved in FoxO1 degradation, we identified a conserved UPR element (UPRE) nucleotide sequence—the prototypical binding site of Xbp1s—in the promoter region of the E3 ubiquitin ligase STUB1 (STIP1 homology and U-Box-containing protein 1) (Fig. 4a). To pursue this, we engineered a luciferase reporter vector harboring the UPRE of STUB1 (STUB1-Luc). Infection of

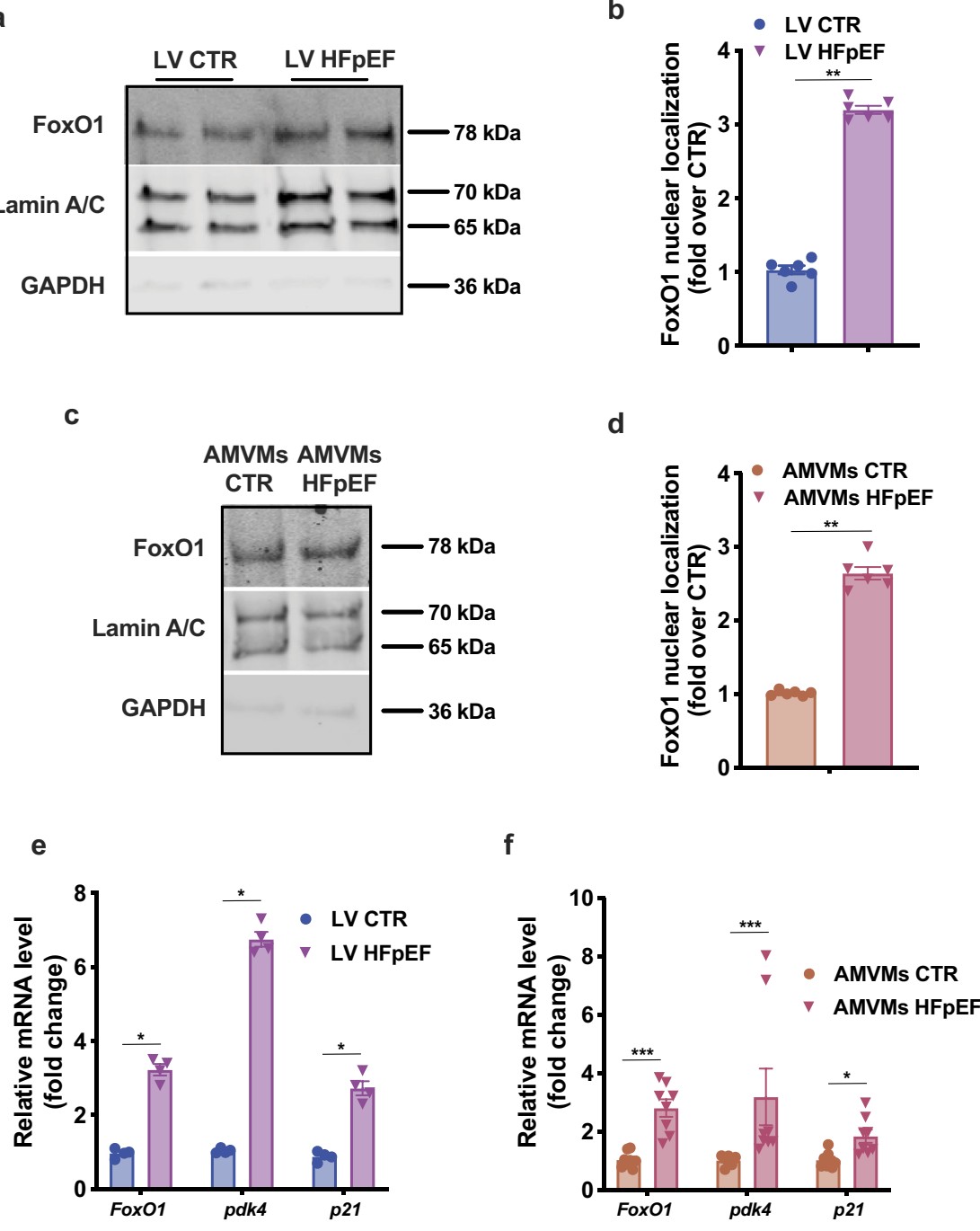

**Fig. 2 FoxO1 activation in hearts and cardiomyocytes from HFpEF mice. a** Representative immunoblot images of FoxO1, Lamin A/C, and GAPDH proteins in nuclear extracts from LV of CTR and HFpEF mice. Images are representative of six hearts/group. **b** Bar graphs depicting intensities of the nuclear FoxO1 protein band ($n = 6$/group). **c** Representative immunoblot images of FoxO1, Lamin A/C, and GAPDH proteins in nuclear extracts from adult mouse ventricular myocytes (AMVMs) of CTR and HFpEF mice. Images are representative of six hearts/group. **d** Bar graphs depicting the nuclear FoxO1 protein band intensities ($n = 6$/group). **e** mRNA levels of *FoxO1*, *pdk4*, and *p21* in LV of CTR and HFpEF mice ($n = 4$/group). **f** mRNA levels of *FoxO1*, *pdk4*, and *p21* in AMVMs of CTR and HFpEF mice ($n = 8$/group). Results are presented as mean ± S.E.M. In **b**, **d**, **e**, and **f** *$p < 0.05$, **$p < 0.005$, ***$p < 0.0005$, unpaired, two-tailed Kolmogorov–Smirnov test.

AdXbp1s in STUB1-Luc-transfected HEK293 cells resulted in concentration-dependent induction of luciferase activity (Fig. 4b). A similar increase in STUB1-luciferase activity was observed when HEK293 cells and NRVMs were treated with the canonical UPR activator tunicamycin (TUN) (Supplementary Fig. 7a, b). Chromatin immunoprecipitation (ChIP) assay confirmed the binding of Xbp1 to the STUB1 promoter (Fig. 4c, d). In addition,

pharmacological inhibition of IRE1α endoribonuclease activity with MK-3946 or Xbp1s knockdown using Xbp1s-specific siRNA abolished TUN-induced increases in STUB1-luciferase activity (Supplementary Fig. 7a, b), confirming the specificity of the Xbp1s-dependent increase in STUB1-luciferase activity.

To corroborate our findings, we infected NRVMs with AdXBP1s or treated them with TUN, observing significant increases in

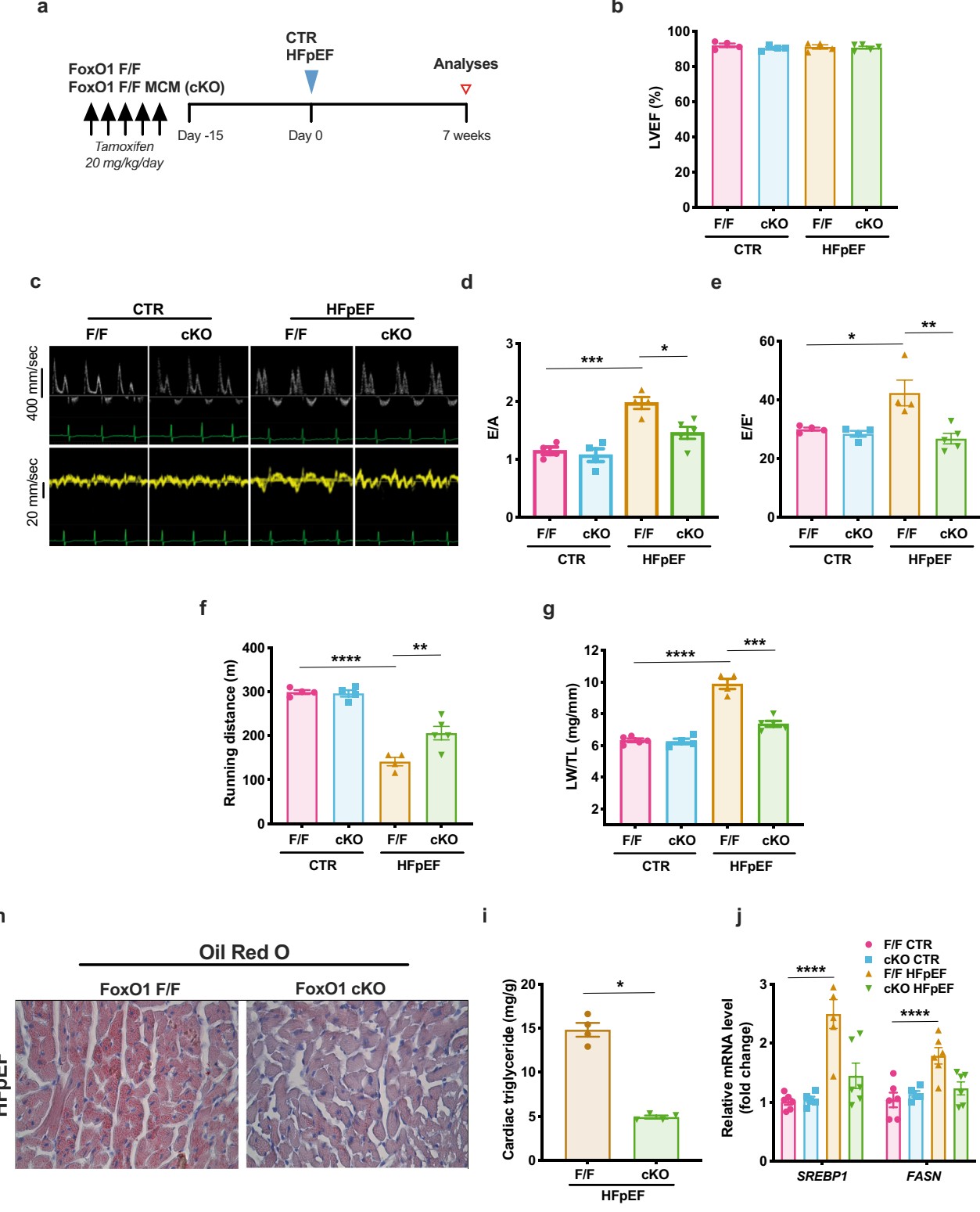

STUB1 mRNA (Fig. 4e and Supplementary Fig. 7c) and protein (Fig. 4f, g) in an Xbp1s-dependent manner (Supplementary Fig. 7c); similar findings were observed in XBP1s TG hearts (Fig. 4h, i). Specificity of the putative STUB1 protein band in cardiomyocytes was confirmed in NRVMs transfected with STUB1-specific siRNA. Importantly, levels of STUB1 mRNA tracked with levels of Xbp1s in vivo, with STUB1 transcript levels manifesting a significant decrease in HFpEF hearts (Fig. 4j). Accordingly, under conditions

of HFD feeding in which Xbp1s is induced, STUB1 levels also increase (Supplementary Fig. 7d) Collectively, these data provide compelling evidence that STUB1 is a direct transcriptional target of Xbp1s.

**STUB1 regulates FoxO1 stability in cardiomyocytes.** Our data suggest that Xbp1s activates STUB1 transcription, promoting the

**Fig. 3 Cardiomyocyte-restricted deletion of FoxO1 improves HFpEF phenotype in mice and reduces cardiac lipid accumulation. a** Experimental design. FoxO1 *flox/flox* (FoxO1 F/F) and FoxO1 *flox/flox/α-MHC-MerCreMer* mice (FoxO1 F/F MCM; cKO) were injected once a day for 5 consecutive days. After 15 days, FoxO1 F/F and FoxO1-cKO were exposed to CTR or HFpEF combination diet (blue triangle). After 7 weeks, mice were subjected to functional analysis and tissue harvesting (red empty triangle). **b** Left ventricular ejection fraction % (LVEF%) of different experimental groups ($n = 4$ for F/F CTR, F/F cKO, and F/F HFpEF groups; $n = 5$ for cKO HFpEF group). **c** Representative pulse wave Doppler (top) and tissue Doppler (bottom) tracings from different experimental groups. Images are representative of four hearts/group. **d** E/A ratio of different experimental cohorts ($n = 4$ for F/F CTR, F/F cKO, and F/F HFpEF groups; $n = 5$ for cKO HFpEF group). **e** E/E' ratio of different experimental cohorts ($n = 4$ for F/F CTR, F/F cKO and F/F HFpEF groups; $n = 5$ for cKO HFpEF group). **f** Running distance during exercise exhaustion test ($n = 4$ for F/F CTR, F/F cKO, and F/F HFpEF groups; $n = 5$ for cKO HFpEF group). **g** Ratio between lung weight (LW) immediately after mouse euthanasia (wet) and tibia length (LW/TL) ($n = 4$ for F/F cKO and F/F HFpEF groups; $n = 5$ for F/F CTR and cKO HFpEF groups). **h** Representative images of Oil Red O staining of LV sections from FoxO1 F/F and FoxO1-cKO HFpEF hearts. Scale bars = 50 μm. Images are representative of four hearts/group. **i** Cardiac triglyceride content in myocardial tissue from FoxO1 F/F and FoxO1-cKO HFpEF hearts ($n = 4$/group). **j** mRNA levels of *SREBP1* and *FASN* in LV of FoxO1 F/F and FoxO1-cKO CTR and HFpEF hearts ($n = 6$ for F/F CTR, F/F HFpEF, and cKO HFpEF groups; $n = 5$ for cKO CTR group). Results are presented as mean ± S.E.M. In **b**, **d–g**, **j** *$p < 0.05$, **$p < 0.005$, ***$p < 0.0005$, ****$p < 0.0001$, two-way ANOVA plus Sidak's multiple comparisons test. **i** ****$p < 0.0001$, unpaired, two-tailed Kolmogorov–Smirnov test.

proteolytic degradation of FoxO1. To examine this more directly, we measured the impact of STUB1 on the half-life of FoxO1 protein. NRVMs were treated with cycloheximide to block protein translation, and steady-state levels of FoxO1 over the course of 8 h were measured. Loss of STUB1 led to a significant decrease in the early rate of decay of FoxO1 and increased the protein's half-life in this assay from ~4 h to almost 8 h. (Fig. 5a, b). In support of this, overexpression of STUB1 in NRVMs using a recombinant adenovirus driving expression of STUB1 (AdSTUB1), induced a decrease in FoxO1 protein levels coupled with an increase in protein ubiquitination as compared with AdGFP-infected control cells (Supplementary Fig. 8a, b). Cardiomyocyte infection with AdSTUB1 did not elicit changes in FoxO1 mRNA levels as compared with AdGFP-infected cells (Supplementary Fig. 8c). In aggregate, these results confirm that STUB1 substantially participates in the regulation of FoxO1 protein stability in cardiomyocytes.

To delineate the role of STUB1 in Xbp1s-dependent control of FoxO1 in cardiomyocytes, we evaluated FoxO1 mRNA and protein levels in STUB1-depleted NRVMs in the setting of Xbp1s overexpression. STUB1 knockdown in cardiomyocytes did not affect FoxO1 transcript levels under basal conditions or after Xbp1s overexpression (Fig. 5c). Strikingly, STUB1 depletion significantly restored FoxO1 protein levels in AdXbp1s-infected NRVMs (Fig. 5d, e). A similar pattern was observed using TUN as an inducer of Xbp1s. TUN-treated NRVMs displayed a reduction in FoxO1 protein, which was significantly reduced by STUB1 knockdown (Supplementary Fig. 9a, b). Similar to our observations with Xbp1s overexpression, TUN treatment in NRVMs increased STUB1 mRNA levels (Supplementary Fig. 9c). In addition, inhibition of Xbp1s splicing with MK-3946 was sufficient to alleviate TUN-induced FoxO1 protein degradation, confirming that TUN induces the reduction of FoxO1 protein levels in an Xbp1s-dependent manner (Supplementary Fig. 9d, e). Of note, neither TUN nor MK-3946 had any effects on FoxO1 mRNA levels in NRVMs (Supplementary Fig. 9f).

Finally, to further confirm the specificity of the Xbp1s–STUB1 interaction in the regulation of FoxO1 stability, we evaluated another FoxO1 E3 ubiquitin ligase, SKP2 (S-Phase Kinase Associated Protein 2)[20]. Whereas infection of NRVMs with AdXBP1s resulted in increases in SKP2 mRNA (Supplementary Fig. 10a), SKP2 knockdown using SKP2-specific siRNA (siSKP2) did not relieve Xbp1s-dependent FoxO1 protein degradation (Supplementary Fig. 10b). Similarly, neither autophagy activation nor inhibition of intracellular proteases alleviated Xbp1s-dependent FoxO1 protein degradation (Supplementary Fig. 10c) Collectively, these data reveal that Xbp1s-dependent degradation of FoxO1 protein in cardiomyocytes is, at least in part, dependent on STUB1.

## Discussion

The global spread of obesity and metabolic dysfunction, coupled with widespread growth in hypertension, are altering the face of cardiovascular disease, an evolution that is likely to continue to accelerate. As a major part of this, HFpEF, a disorder devoid of evidence-based therapy, is increasing in prevalence. Here, we employed a clinically relevant murine model of cardiometabolic HFpEF and made a series of novel mechanistic observations: (1) Xbp1s, which is decreased in HFpEF[9], reduces myocardial lipid accumulation; (2) FoxO1 promotes lipid accumulation in HFpEF cardiomyocytes; (3) Xbp1s promotes FoxO1 protein degradation; (4) cardiomyocyte-specific deletion of FoxO1 ameliorates the HFpEF phenotype and diminishes cardiac steatosis; (5) the E3 ubiquitin ligase STUB1, a direct transcriptional target of Xbp1s, is also downregulated in HFpEF and governs FoxO1 stability in cardiomyocytes. In aggregate, we unveil the role of FoxO1 as a driver of lipid accumulation in cardiomyocytes, functioning under the control of its molecular brake, Xbp1s, and emerging as a critical contributor to HFpEF pathogenesis. We propose a novel pathophysiological model of HFpEF in which suppression of the Xbp1s arm of the UPR blunts STUB1 expression, leading to the stabilization and overactivation of FoxO1 in cardiomyocytes with consequent cardiomyocyte steatosis and associated detrimental sequelae (Fig. 6).

Obesity and diabetes significantly increase the risk of atherosclerotic coronary artery disease and myocardial infarction, which almost inevitably impairs systolic function leading to HFrEF. However, obese people are at markedly increased risk of HFpEF independent of the occurrence of ischemic cardiac injury[21,22]. Indeed, metabolic disorder-related HFpEF is the most prevalent phenotype of HFpEF in the community[3,23]. Prior work has described fatty infiltration of cardiomyocytes in failing hearts, causally linking cardiac lipid overload with systolic and diastolic dysfunction[24–26]. Thus, HFpEF can be considered the most common clinical manifestation of lipotoxic cardiomyopathy.

It is important to note that HFpEF, both clinically and mechanistically, does not completely overlap with obesity, and that obesity is one—the most common—of the many comorbidities shaping the complexity of the pathophysiological mechanisms of HFpEF. Accordingly, in obese (HFD-fed) mice, we observed activation, rather than suppression, of Xbp1s[9,10] and STUB1, coupled with activation of FoxO1, suggesting a feedback circuit whereby Xbp1s acts to increase transcription of STUB1 to blunt the persistent activation of FoxO1 observed under HFD conditions. Conversely, in HFpEF hearts, downregulation of Xbp1s leads to suppression of STUB1 transcription, further increasing FoxO1 activation and increasing cardiomyocyte lipid accumulation. Therefore, as obesity alone does not elicit HFpEF, the mechanistic underpinnings of obesity-induced cardiomyopathy are different from those in HFpEF,

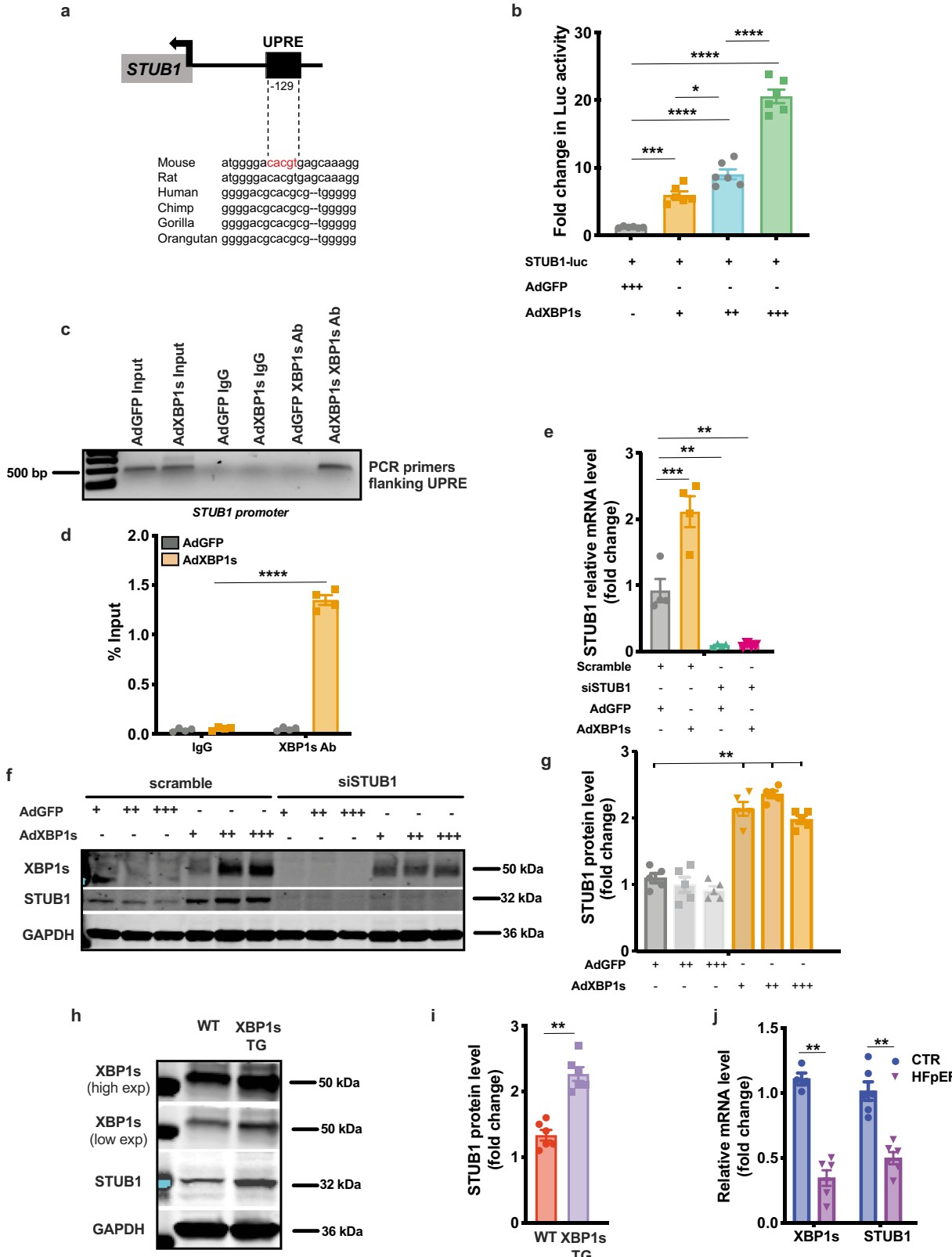

supporting a model in which FoxO1 overactivation in HFpEF leads to the detrimental functional consequences of cardiomyocyte lipid accumulation.

The precise mechanisms whereby lipids induce cardiac dysfunction are unclear. In principle, pathological lipid accumulation in cardiomyocytes can result from increased FA uptake/

biogenesis, decreased FA oxidation, or a combination of the two. In the hearts of obese humans and rodents, an increase, rather than a decrease, in FA oxidation has been reported[27,28]. Conversely, impaired FA oxidation is a hallmark of metabolic dysfunction in HF. In patients with HFrEF without metabolic abnormalities (i.e., diabetes and obesity) only a minimal increase

Fig. 4 STUB1 is a direct transcriptional target of Xbp1s. a Conserved consensus sequence of Xbp1s-binding site (unfolded protein response element—UPRE) in the promoter region of STUB1. 129 bp from the transcriptional start site (arrow). b Luciferase (Luc) activity in human embryonic kidney 293 cells (HEK293) transfected with STUB1-luciferase reporter construct (STUB1-Luc) and transduced with the increasing multiplicity of infection of AdGFP or AdXbp1s ($n = 6$ biologically independent experiments). c Electrophoretic analysis of chromatin immunoprecipitation (ChIP) assay of STUB1 promoter in NRVMs transduced with AdGFP or AdXbp1s. ChIP was performed with either control mouse immunoglobulin G (IgG) or anti-Xbp1s antibody. PCR was conducted with primers spanning the UPRE site. Images are representative of 4 independent experiments. d Densitometric analysis of STUB1 ChIP band intensities in the different experimental groups ($n = 4$ biologically independent experiments). e mRNA level of *STUB1* in NRVMs transfected with STUB1-specific small interfering RNA (siSTUB1) or scrambled siRNA control and transduced with AdGFP or AdXbp1s ($n = 4$ biologically independent experiments). f Representative immunoblot images of Xbp1s, STUB1 and GAPDH proteins from NRVMs transfected with siSTUB1 or scrambled siRNA control and transduced with increasing multiplicity of infection of AdGFP or AdXbp1s. Images are representative of five independent experiments. g Densitometric analysis of STUB1 protein band in the different experimental groups ($n = 5$ biologically independent experiments). h Representative immunoblot images of Xbp1s, STUB1, and GAPDH proteins from LV of WT and Xbp1s TG mice. High and low exposure (high/low exp). Images are representative of six hearts/group. i Bar graphs depicting STUB1 protein band intensities in WT and Xbp1s TG LV samples ($n = 6$/group). j mRNA level of *XBP1s* and *STUB1* in LV of CTR and HFpEF hearts ($n = 4$ for Xbp1s CTR group; $n = 6$ for the remaining groups). Results are presented as mean ± S.E.M. In b and e, *$p < 0.05$, **$p < 0.005$, ***$p < 0.0005$, ****$p < 0.0001$, one-way ANOVA plus Sidak's multiple comparisons test. In d, g, i, j **$p < 0.005$, ****$p < 0.0001$, unpaired, two-tailed Kolmogorov–Smirnov test.

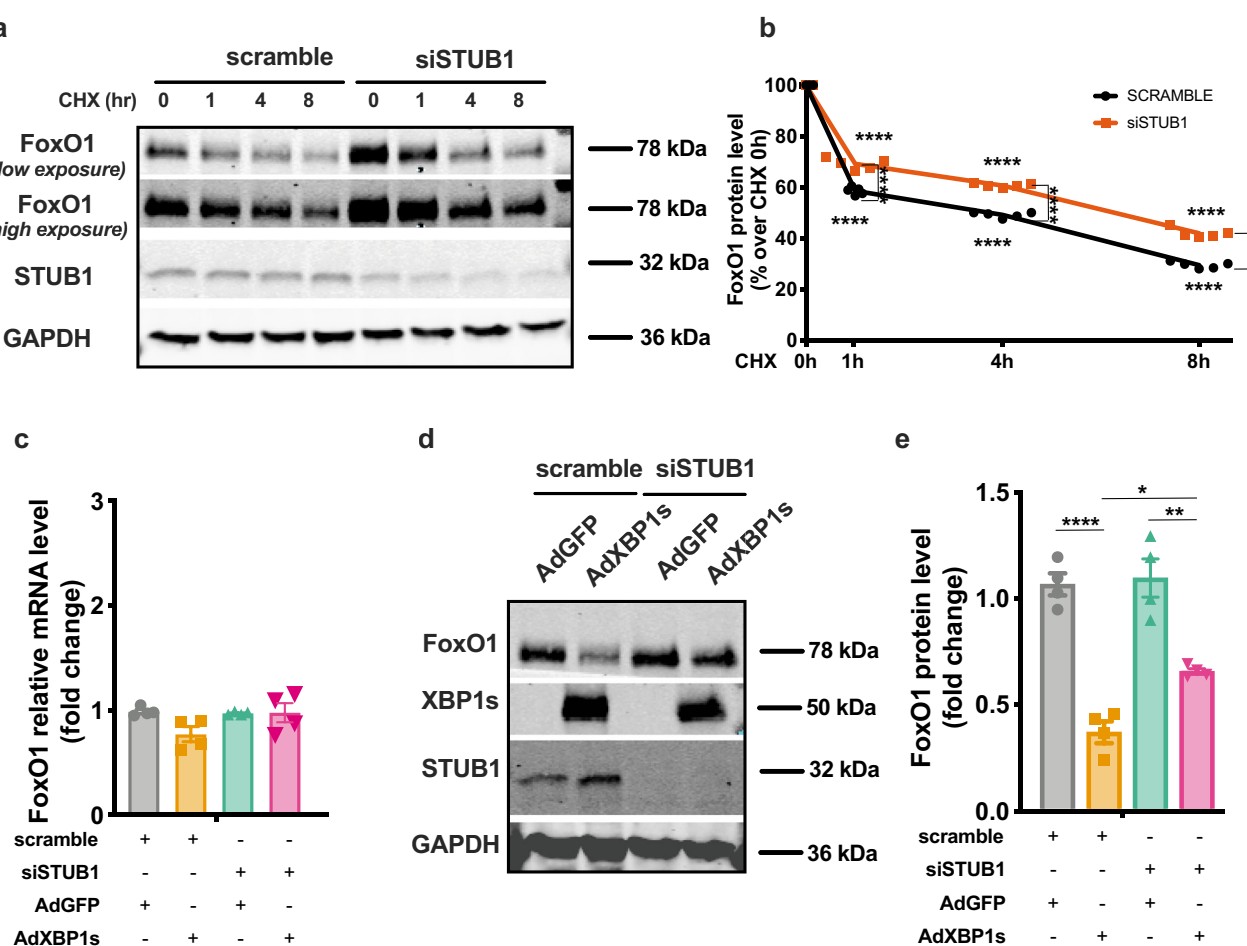

Fig. 5 STUB1 mediates Xbp1s-dependent FoxO1 protein degradation. a Representative immunoblot images of FoxO1, STUB1, and GAPDH proteins from NRVMs transfected with siSTUB1 or scrambled siRNA control and treated with cycloheximide (CHX) for different durations (0, 1, 4, and 8 h). Images are representative of five independent experiments. b Densitometric analysis of FoxO1 protein band in the different experimental groups. ($n = 5$ biologically independent experiments). c mRNA level of *FoxO1* in NRVMs transfected with siSTUB1 or scrambled siRNA control and transduced with AdGFP or AdXbp1s ($n = 4$ biologically independent experiments). d Representative immunoblot images of FoxO1, Xbp1s, STUB1, and GAPDH proteins from NRVMs transfected with siSTUB1 or scrambled siRNA control and transduced with AdGFP, AdXbp1s. Images are representative of four independent experiments. e Densitometric analysis of FoxO1 protein band in the different experimental groups. ($n = 4$ biologically independent experiments). Results are presented as mean ± S.E.M. In b, c, and e *$p < 0.05$, **$p < 0.005$, ****$p < 0.0001$, two-way ANOVA plus Sidak's multiple comparisons test.

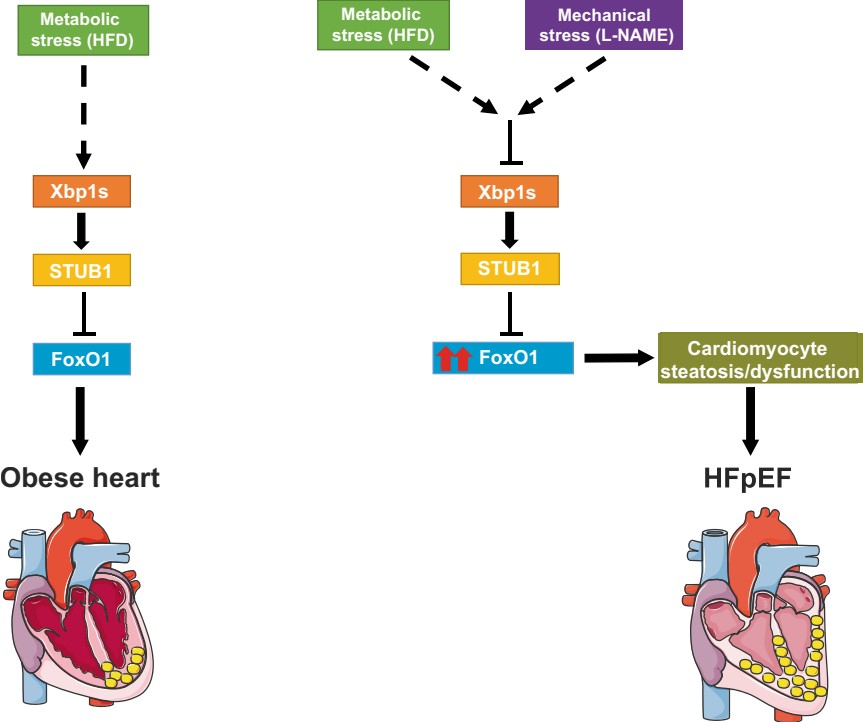

**Fig. 6 Proposed model.** Schematic depicting a mechanistic model of Xbp1s-STUB1-FoxO1 axis in HFpEF.

in myocardial triglyceride content has been observed[25]. Conversely, in obese/diabetic patients with HFrEF, significant accumulation of intramyocardial lipid has been reported[25]. These data suggest that impaired FA oxidation alone is insufficient to elicit significant myocardial steatosis, and cardiac lipid overload is largely a function of increased FA uptake/biogenesis. Indeed, although increased FA oxidation has long been considered a culprit contributing to cardiac dysfunction in obese models, recent data suggest that increased FA oxidation does not cause cardiac dysfunction in mice[29,30]. Taken together, these observations suggest that cardiac lipotoxicity is not attributable to increased fatty-acid oxidation per se, but rather to an imbalance of fatty-acid supply, storage, and use. Consistent with this, we have observed an increase in cardiac lipogenic signature coupled with reduction in mitochondrial FA oxidation in cardiometabolic HFpEF. Similarly, in two models of reduced cardiac lipid accumulation in HFpEF (Xbp1 overexpression and FoxO1 deletion) we did not observe an increase in FA oxidation in cardiomyocytes, suggesting that the prevalent mechanisms of lipid accumulation in HFpEF cardiomyocytes are activation of FA uptake/biogenesis pathways.

Dysregulation of the UPR pathway has emerged as a hallmark of cardiac alterations in HFpEF[31,32]. We have previously demonstrated, in both clinical and experimental HFpEF, inflammation-dependent suppression of the IRE1α/Xbp1s arm of the UPR[9]. Restoration of Xbp1s levels in failing cardiomyocytes results in diminished diastolic dysfunction and amelioration of the global HFpEF phenotype[9]. Here, we have unveiled proteasomal degradation of FoxO1 as a major mechanism of Xbp1s-dependent cardioprotective effects in HFpEF.

The interplay between the UPR and lipid biology is complex. FA can activate the UPR, and specific branches of the UPR regulate lipid metabolism[33]. For example, in hepatocytes, Xbp1s regulates lipogenesis[34,35], a function that is independent of its classical role in ER stress and critically involves FoxO1[15]. With respect to cardiac biology, FoxO1 participates in remodeling, autophagy, apoptosis, responses to oxidative stress, and

regulation of metabolism[16,36,37]. In addition, FoxO1 has emerged as an important target of insulin and other growth factors in the myocardium[16], and we have reported that FoxO1 is persistently activated in cardiac tissue from murine models of long-term diabetes[17]. Despite this, upstream regulators of FoxO1-dependent regulation of lipid homeostasis in cardiomyocytes remain largely unknown, and a role of FoxO1 in HFpEF has not been reported previously.

Xbp1s is a powerful transcription factor acting through a variety of transcriptional targets involved in many fundamental cellular responses to stress. We report here that the E3 ubiquitin ligase STUB1 is a direct downstream target of Xbp1s. STUB1 is a major FoxO1 ubiquitin ligase driving its proteasomal degradation[38]. However, whereas we observed STUB1-dependence of FoxO1 protein degradation upon Xbp1s activation, we recognize that this may not be the only mechanism involved in FoxO1 degradation upon metabolic stress in cardiomyocytes; other mechanisms may play a role. STUB1, also known as carboxyl terminus of Hsc70-interacting protein (ChIP), is a dual function co-chaperone/ubiquitin ligase that is highly expressed in the heart and vasculature[39]. Interestingly, a cardioprotective role of STUB1 involving, at least in part, the modulation of lipid accumulation, has been observed in two models of HFrEF, suggesting a central role of STUB1 in cardiomyocyte homeostasis[40,41]. Direct targeting of transcription factors for therapeutic purposes has proven challenging[42]. Therefore, the identification of negative upstream regulators of FoxO1 such as STUB1 might result in valuable therapeutic targets to diminish FoxO1 activation and its negative metabolic consequences. Indeed, strategies aiming to enhance STUB1 functionality for therapeutic applications in cardiovascular disease have been proposed recently[43], supporting the potential feasibility of targeting STUB1 for clinical gain.

We have demonstrated that FoxO1 suppression in HFpEF is beneficial. Interestingly, these beneficial effects occur in the absence of a significant reduction in cardiac hypertrophy. By contrast, we recently described a pro-hypertrophic role for FoxO1 acting through cardiomyocyte thyroid hormone metabolism[37].

Together, these data suggest that FoxO1 plays different roles in cardiac pathology depending on the specific stress conditions.

Another major metabolic abnormality observed in HF is systemic insulin resistance. HF causes insulin resistance[44] and insulin resistance worsens HF, a feedback regulatory circuit contributing to HF susceptibility and progression. We and others have positioned FoxO1 as a pivotal element in the control of insulin signaling in the heart. Forced expression of FoxO1 in cardiomyocytes culminates in reduced insulin sensitivity and impaired glucose metabolism[16]. In HFpEF hearts, we observe increased activity of PDK4, which will repress glucose oxidation[45]. Indeed, an increase in PDK4 levels is a common feature of hearts subjected to metabolic stress. Interestingly, in cardiometabolic HFpEF the increase in PDK4 is independent of FoxO1.

Beyond glucose metabolism, less well-known is the role of FoxO1 in regulating cardiac lipid homeostasis. FoxO1 nuclear compartmentalization and enhancement of its transcriptional activity, together with increases in FA uptake, have been observed in models of long-term diabetes and obesity[46,47]. Despite this, debate regarding the functional consequences of persistent diabetes or obesity on cardiac function persists, with some reporting little or no effect[48,49] and others reporting evolution toward HFrEF[26,50]. Here, we have provided the clinical context—the syndrome of HFpEF—in which FoxO1 overactivation promotes cardiac lipotoxicity. We observed impaired cardiac mitochondrial FA oxidation in HFpEF. Interestingly, in FoxO1-cKO mice subjected to HFpEF, we observed a trend toward amelioration of mitochondrial fatty-acid oxidation in the absence of changes in glucose oxidation. Collectively, these results imply that a major disruption of energetic substrate utilization occurs in HFpEF, and the downregulation of FoxO1 tends to mitigate this energetic disturbance, acting through a decrease in HFpEF-induced lipotoxicity.

It is important to note that cardiometabolic alterations in HFpEF, and their underlying pathophysiological mechanisms, exhibit a certain degree of sex dependence. Female mice seem to be protected from experimental HFpEF[10], whereas some evidence suggests a predilection for HFpEF in women[3,51]. Despite this, sexual dimorphism in HFpEF pathophysiology is an under-developed area of investigation. Therefore, elucidation of sex-neutral signaling pathways involved in HFpEF pathogenesis is warranted.

Here, we have unveiled an Xbp1s-FoxO1 signaling circuit in diseased cardiomyocytes, thereby defining events downstream of the UPR in heart that govern metabolic events. We have identified a biological axis, Xbp1s-STUB1-FoxO1, in the regulation of cardiac steatosis in HFpEF, providing a novel pathophysiological model for cardiometabolic HFpEF. The field of experimental HFpEF is rapidly moving from a simplistic view in which HFpEF equals diastolic dysfunction to a view of HFpEF as a complex syndrome in which metabolic alterations contribute importantly to its pathogenesis. As such, delineation of alterations in lipid metabolism and their consequences on cardiac function is of paramount importance in deciphering mechanisms underlying this prevalent and devastating syndrome.

## Methods

**Experimental animals**. All experiments involving animals conformed to the Guide for the Care and Use of Laboratory Animals published by the US National Institutes of Health (NIH Publication 8th edition, update 2011) and were approved by the Institutional Animal Care and Use Committee of the University of Texas Southwestern Medical Center. All studies were in compliance with all ethical regulations. C57BL/6 N mice were used for wild-type (WT) studies. Tetracycline responsive elements (TRE)-Xbp1s mice were crossed with mice harboring a tetracycline transactivator (tTA) transcription factor driven by α-myosin heavy chain promoter (αMHC-tTA) to generate mice with cardiomyocyte-specific inducible overexpression of Xbp1s (Xbp1s TG)[9,52]. Cardiomyocyte-specific FoxO1 knockout

(cKO) animals, mice harboring a floxed FoxO1 allele were crossed with α-MHC-MerCreMer transgenic mice. FoxO1-cKO and corresponding control floxed or α-MHC-MerCreMer mice were maintained on an FVB genetic background. Male adult (10/12-week-old) mice were used in the experiments. Mice were maintained on a 12-hour light/dark cycle from 6 AM to 6 PM, 65–75 °F, 40–50% humidity and had unrestricted access to food (#2916, Teklad for CHOW groups and #D12492, Research Diet Inc. for the HFD groups) and water. N[w]-nitro-l-arginine methyl ester (L-NAME; 0.5 g/L, Sigma-Aldrich) was supplied in the drinking water or embedded in the chow (custom made #D16082402, Research Diet Inc). HFpEF was induced by feeding mice with HFD + L-NAME diet from 5 to 15 weeks[9,10]. Pregnant Sprague-Dawley rats were purchased (Charles River Laboratories) and were used for the sole purpose of harvesting primary cardiomyocytes from 1–2-day old rat pups.

**Conventional echocardiography and Doppler imaging**. Transthoracic echocardiography was performed using a VisualSonics Vevo 2100 system equipped with MS400 transducer (Visual Sonics Inc). Indices of systolic function were obtained in conscious, gently restrained mice from M-mode scans at midventricular level. Indices of diastolic function were obtained in anesthetized mice from apical four-chamber view using pulsed-wave and tissue Doppler imaging. Anesthesia was induced by 5% isoflurane and, during echocardiogram acquisition, isoflurane was reduced to 1.0–1.5% and adjusted to maintain a heart rate >400 beats per min[9,10]. All parameters were measured at least 3 times, and averages are presented.

**Exercise exhaustion test**. Exercise tests were performed as depicted in the figures using the following protocol. After acclimatization, mice ran uphill (20°) on a treadmill apparatus (Columbus Instruments) according to a ramping speed protocol[9,10] until exhaustion. Exhaustion was defined as the inability of the mouse to return to running within 10 s of direct contact with an electric-stimulus grid.

**Histology and myocardial lipid measurements**. Hearts were harvested and flash frozen in an embedding medium containing a 3:1 mixture of Tissue Freezing Medium (Triangle Biomedical Sciences, Durham, NC) and gum tragacanth (Sigma, St. Louis, MO). 0.3% Oil Red O (BDH, Poole, United Kingdom) staining was performed on 5 μm-thick sections according to standard procedures[53]. Cardiac sections were visualized with a Leica DM2000 upright photomicroscope. Total triglyceride content was determined biochemically from myocardial tissue collected immediately after euthanasia. Colorimetric quantification of total triglycerides was performed using Triglycerides Reagent—Thermo Fisher Scientific according to the manufacturer's instructions[17,54].

**Cardiomyocyte isolation, treatment, transfection, and adenovirus production, and transduction**. AMVMs were isolated from littermates of each experimental group as follows[9]. Hearts were dissected and immediately digested by retrograde perfusion with perfusion buffer (113 mM NaCl, 4.7 mM KCl, 0.6 mM KH₂PO₄, 0.6 mM Na₂HPO₄, 1.2 mM MgSO₄, 10 mM HEPES, 12 mM NaHCO₃, 10 mM KHC0₃, 30 mM taurine, 10 mM butanedione monoxime, and 5.5 mM glucose) for 1 min followed by buffer containing liberase TM (0.025 mg mL⁻¹, Roche) and trypsin (0.025%) for 14 min. The left ventricle was dissected and minced in a perfusion medium supplemented with 10% dialyzed fetal bovine serum. After filtration, AMVMs were allowed to sediment by gravity and non-cardiomyocytes were discarded. Calcium reintroduction was performed stepwise from 0 to 1.8 mM in six steps. AMVMs were stained with LipidTOX deep red (1:200 dilution; Invitrogen) for 1 h according to the manufacturer's instructions. Images were obtained using a Zeiss LSM 880 upright confocal fluorescence microscope. NRVMs were isolated from 1 to 2-day-old Sprague-Dawley rats[9]. NRVMs were cultured in Dulbecco's modified Eagle's medium/M199 (3:1) containing 3% fetal bovine serum and antibiotics before performing the experiments. NRVMs were treated in serum-free medium with BSA/oleic acid complexes (Sigma-Aldrich, O3008-5ML, 2 moles of oleic acid per 1 mole of BSA) at the concentration of 200 μM for 3 h, 6 h, 24 h or 48 h. Free fatty-acid BSA (Sigma-Aldrich, A8806-5G) was used as control. Neutral lipids were visualized by Oil O Red staining as described above. For gene knockdown, NRVMs were transfected with three sequence-independent, STUB1-, FoxO1-, and siSKP2 (Mission, Sigma-Aldrich) using Lipofectamine RNAiMax (Invitrogen) in Opti-MEM (Gibco). After 6 h, cells were washed twice with growth medium and culture overnight in growth medium. For adenovirus-mediated protein overexpression, cells were transduced for 6 h with AdXBP1s[52], AdcaFoxO1[37], and AdSTUB1 adenoviruses with increasing multiplicity of infection (MOI) of virus, and cells were harvested 24-hour post-infection. A GFP adenovirus (AdGFP) construct was used as control at the same MOI. Human STUB1 cDNA was obtained from UT Southwestern Center for Human Genetics and subcloned into an adenovirus expression vector using Adeno-X Adenoviral System 3 (Takara) following the manufacturer's instructions. Adeno-X 293 cell line was purchased from Clontech (Adeno-X 293 Cell Line, 632271). When appropriate, NRVMs and HEK293 cells were treated with MG132 (Sigma-Aldrich, 10μM; 4 h), bafilomycin A1 (LC Laboratories, 50 nM; 2 h), cell-permeable protease inhibitor cocktail (Sigma-Aldrich, 1:200 dilution; 2 h), TUN (Sigma-Aldrich, 10 μg/mL; 24 h), MK-3946 (Tocris, 10 μM, 24 h), cycloheximide (Sigma-Aldrich, 5 μg/mL, 1–8 h).

**Lactate dehydrogenase (LDH) assay**. To evaluate cell survival, LDH release was quantified using the CytoTox96 cytotoxicity kit (Promega) according to the manufacturer's instructions. Each experiment was performed in three biological replicates each time in triplicate. LDH release was calculated as follows: (medium LDH)/(medium LDH + intracellular LDH).

**Protein extracts, subcellular fractionation, and immunoblot analysis**. Whole protein extracts from frozen mouse hearts, AMVMs and NRVMs were prepared by lysis in ice-cold modified radio immunoprecipitation assay (RIPA) buffer (150 mM NaCl, 50 mM Tris HCl pH 7.4, 1% Triton-X 100, 0.5% sodium deoxycholate, 0.1% SDS, 5 mM ethylenediaminetetraacetic acid (EDTA), 2 mM EDTA) containing protease and phosphatase inhibitors. Nuclear and cytosolic extracts were prepared using the NE-PER™ Nuclear and Cytoplasmic Extraction Kit (Thermo Scientific) following the manufacturer's instructions. Proteins were separated by sodium dodecyl sulfate-polyacrylamide gel electrophoresis on 4–20% gradient gels (Bio-Rad) and transferred to nitrocellulose membranes. An Odyssey scanner (LI-COR version 3.0) was used as detection system. The separation between the nuclear and cytosolic fractions was verified by blotting for the cytosolic protein GAPDH and the nuclear membrane protein Lamin A/C. Tandem ubiquitin binding entities for the detection of ubiquitinated protein were used according to the manufacturer's instructions (Life Sensors). Proteins were detected with a 1000-fold dilution of the following primary antibodies: FoxO1 (#2880, Cell Signaling); FoxO3 (#2497, Cell Signaling); Xbp1s (poly6195, BioLegend); GFP (AB10145, Sigma-Aldrich); STUB1 (#2080, Cell Signaling); Ubiquitin (#3933, Cell Signaling); LC3 (previously developed[55]); Lamin A/C (05-714, Sigma-Aldrich); PDK4 (previously developed[56]), OXPHOS cocktail (ab110413, Abcam). 10,000-fold dilution of GAPDH (10R-G109a, Fitzgerald). Uncropped scans of immunoblot images are available in the Supplementary Information.

**RNA isolation and qPCR**. Total RNA was extracted from murine hearts, AMVMs, or NRVMs using TRIzol reagent or Quick-RNA™ MicroPrep kit (Zymo Research). A total of 500 ng RNA was used for reverse transcription using iScript reagent (Bio-Rad). qPCR reactions were performed in triplicate with SYBR master mix (Bio-Rad). The $2^{-\Delta\Delta CT}$ relative quantification method, using 18 S for normalization, was used to estimate the amount of target mRNA in samples, and fold ratios were calculated relative to mRNA expression levels from control samples. qPCR primer sequences are provided in Supplementary Table 2.

**Analysis of mitochondrial respiratory function in hearts and cardiomyocytes**. NRMVs were transduced with AdGFP and AdXbp1s as described above. Cardiac mitochondria were isolated by ultracentrifugation in ice-cold isolation buffer (210 mM MOPS, 70 mM mannitol, 10 mM sucrose, 1 mM EDTA, pH = 7.4). Oxygen consumption was measured using a fluorescence-based oxygen sensor (NeoFox, Ocean Optic) connected to a phase measurement system from the same company. The sensor was calibrated according to the manufacturer's instructions[57]. NRVM cell numbers were counted using a hemocytometer. In brief, $1 \times 10^4$ cardiomyocytes were incubated in a solution containing KMES 60 mM, MgCl2 223 mM, KH2PO4 10 mM, Hepes 20 mM, taurine 20 mM, mannitol 110 mM, EGTA 0.5 mM, and DTT 0.3 mM (pH 7.1) at 4 °C, and permeabilized with saponin (50 ng/mL) for 5 min before substrates were added. Isolated mitochondria from hearts (0.25 mg/mL) were incubated with the following buffer: 210 mM MOPS, 70 mM mannitol,10 mM sucrose, 5 mM KH2PO4, pH = 7.4[58]. To measure complex I mediated OCR, glutamate (10 mM) and malate (5 mM) were added. State3 respiration was activated by adding ADP (2.5 mM), whereas state 4 respiration was measured after adding oligomycin (1 μM). For complex II-mediated OCR, 5 mM succinate was added. For the measuring of palmitoyl/carnitine-mediated OCR, 25 μM palmitoyl-carnitine was added to each experiment.

**Luciferase and chIP assays**. Transcriptional assays using pDR4, a luciferase vector harboring two STUB1-UPREs in the presence of AdXbp1s or AdGFP control were performed in HEK293 (Clontech; cat# 632271) and NRVMs[37]. The luciferase vector was transfected into HEK293 cells using Fugene HD (Roche) and into NRVMs using Lipofectamine 3000 (Thermo-Fisher). Luciferase activity was measured using a dual-luciferase kit (Promega) and normalized to renilla activity. Quantitative analyses of ChIP assays were conducted to assess Xbp1s occupancy at the STUB1 promoter using the Zymo-Spin ChIP kit according to the manufacturer's instructions (Epigenetics). In brief, AdGFP- and AdXbp1s-infected NRVMs were exposed to 1% formaldehyde with gentle shaking (15 min, room temperature). After quenching the cross-linking reaction with glycine, cells were washed twice with PBS, and then homogenized in nuclear extraction buffer. The homogenate was centrifuged (10 min, $500 \times g$, 4 °C) and the supernatant was discarded. The nuclear pellet was resuspended in nuclear lysis buffer, sonicated, and centrifuged (15 min, $12,000 \times g$, 4 °C) to prepare chromatin solution. Chromatin solution was diluted 10-fold and incubated with control (IgG) and Xbp1s antibody overnight at 4 °C with rotation. DNA purified from chromatin solutions was used as input. Purified DNA with or without immunoprecipitation was used to determine relative Xbp1s occupancy by comparing the Ct value of 10-fold diluted input DNA with undiluted immunoprecipitated DNA samples in control (IgG) and

Xbp1s antibody. Primers used were (forward, reverse): GGCGGAGCCTTGGTCTGA, CGATCCGCCACCGGAACT).

**Statistical analysis**. Results are presented as mean ± SEMM. Differences were analyzed by two-tailed unpaired Kolmogorov–Smirnov test for experiments with two groups and one-way or two-way analysis of variance plus Sidak's or Tukey's post hoc test for multiple comparisons as appropriate in experiments including ≥3 groups. A minimum value of $p < 0.05$ was considered statistically significant. All experiments were performed with at least three biological replicates. Statistical analyses were conducted using GraphPad Prism software 8.0. No statistical analysis was used to predetermine sample sizes; estimates were made based on our previous experience, experimental approach, availability, and feasibility required to obtain statistically significant results. Experimental animals were randomly assigned to each experimental/control group. Investigators were blinded to the genotypes of the individual animals during the experiments.

**Reporting summary**. Further information on research design is available in the Nature Research Reporting Summary linked to this article.

## Data availability
The authors declare that the data supporting the findings of this study are available within the paper and its Supplementary information files. Any remaining data that support the results of the study will be available from the corresponding author upon reasonable request. Source data are provided with this paper.

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

## Acknowledgements

We thank all members of the Hill laboratory for constructive discussion. We thank members of the Szweda Lab at UT Southwestern for their expertize in mitochondrial biology. We thank Dr. Xiang Luo for isolation of NRVMs. This work was supported by grants from the NIH: HL-120732 (JAH), HL-128215 (JAH), HL-126012 (JAH), HL-147933, (JAH), HL-155765 (JAH), F32HL136151 (KMF), F32HL142244 (DT), American Heart Association (AHA) 16POST30680016 and 19CDA34680003 (FA), 16PRE29660003 (SYK), 14SFRN20510023 (JAH), 14SFRN20670003 (JAH), AHA and the Theodore and Beulah Beasley Foundation grant 18POST34060230 (GGS), Fondation Leducq grant number 11CVD04 (JAH), Cancer Prevention and Research Institute of Texas grant RP110486P3 (JAH) and RP180404 (VGZ) and by Agencia Nacional de Investigacion y Desarrollo (ANID, Chile), FONDAP 15130011 and FONDECYT 1200490 (SL). Images in Fig. 6 were adapted from vectors available at Servier Medical Art (https://smart.servier.com/). Servier Medical Art is licensed under a Creative Commons Attribution 3.0 Unported Licence (https://creativecommons.org/licenses/by/3.0/).

## Author contributions

G.G.S. conceived the project, designed and performed most of the experiments, conducted most of the analyses and wrote the manuscript. F.A., S.Y.K., D.T., A.F., H.P., S.D., X.W., K.M.F, E.V., S.B.S., M.H.W., T.M.H., Y.L., and H.Y. performed experiments and provided the corresponding analyses. N.J. performed experiments and managed mouse colonies. H.I.M supported the mouse work. V.Z., S.L., and T.G.G. contributed to the experimental design and manuscript preparation. J.A.H. conceived the project and contributed to manuscript preparation.

## Competing interests

G.G.S., T.G.G., and J.A.H. are co-inventors on a patent application (PCT/US/2017/037019) that was filed in June 2017 (provisional application filed in June 2016). The patent relates to the diet used for modeling HFpEF. The remaining authors declare no competing interests.
