## [Peer Review File · Nature Communications]

REVIEWER COMMENTS

Reviewer #1 (Remarks to the Author):

This manuscript describes a link between XBP1s and FOXO1 that is involved in regulating cardiac lipid accumulation and function in the context of the HFpEF heart failure model. This link is mediated through the posttranslational proteasomal degradation of FOXO1 through an XBP1s-dependent mechanism. The authors suggest that this degradation is mediated through XBP1s-dependent upregulation of the E3 ligase STUB1.

Overall, this manuscript covers the same XBP1s-dependent regulation of FOXO1 previously shown to regulate lipid metabolism in the liver (see Zhou et al (2011) Nat Med) in the context of cardiac lipid regulation in HFpEF. The XBP1s-FOXO1 link is not new, although its specific importance in the context of HFpEF has not been reported to my knowledge. That being said, there really isn't anything surprising in the link between XBP1s and FOXO1 in the context of lipid regulation in the heart based on published work from this group and many others. The new information in this manuscript comes from the mechanism describing XBP1s-dependent suppression of FOXO1 through XBP1s-dependent upregulation STUB1. The data shown in Figure 4 does seem to suggest that STUB1 is a transcriptional target of XBP1s, although the authors should do the same experiments using other approaches that do not rely on XBP1s overexpression (e.g., treat with tunicamycin +/- an IRE1 inhibitor or Xbp1 siRNA). However, the data in Figure 5 linking STUB1 to FOXO1 degradation is extremely weak. siRNA-dependent knockdown of Stub1 did not significantly change the degradation of FOXO1 induced by XBP1s overexpression in Figure 5B,C. I believe the authors are claiming that the fourth bar in Figure 5C is significantly different than the second bar in Figure C, but, even if they are statistically significant, this is not sufficient a difference to convince that STUB1 is required for XBP1s-dependent FOXO1 degradation. Moreover, the amount of XBP1s overexpression under these two conditions are not the same between these two conditions based on the gel shown in Figure 5B. Ultimately, the data included in this manuscript do not link STUB1 to the XBP1s-dependent degradation of FOXO1 significantly limiting the novelty of this work.

Considering that the main novelty of this paper (i.e., the XBP1s-STUB1 connection and its involvement in FOXO1 degradation) is not sufficiently demonstrated, I find that this manuscript is not ready for publication at this time. It is interesting that the XBP1s-FOXO1 regulatory axis contributes to cardiac function in the context of HFpEF, but in my opinion that link is not sufficient of an advance to merit publication in Nat Comm considering the significant published literature highlighting this same signaling mechanism in other tissues. I encourage the authors to better define the mechanism of the XBP1s-dependent FOXO1 degradation and if it does involve STUB1 to further pursue this relationship with additional experimental efforts that are more convincing than what is included in this manuscript. It is an interesting and important question that just isn't sufficient addressed in the included work.

Reviewer #2 (Remarks to the Author):

This study examined the potential mechanisms involved in the increase in FoxO1 and steatosis in a mouse model of HFpEF. A decrease in Xbp1s in HFpEF is proposed to decrease STUB1, a transcriptional target of Xbp1s. This resulted in a decreased ubiquitination of FoxO1 and an increase in nuclear FoxO1. Over-expression Xbp1s or knocking out FoxO1 resulted in an amelioration of the HFpEF phenotype. In addition, forced expression of Xbp1s in cardiomyocytes resulted in proteosomal degradation of FoxO1 due to activation of the E3 ubiquitin ligase STUB1. It is concluded that the Xbp1s-FoxO1 axis is a pivotal mechanism in the pathogenesis of HFpEF.

General Comments:

Previous studies by the authors have shown that in murine and human HFpEF a down-regulation of Cbp1s correlates with the development of the syndrome. This study adds to this previous observation by showing that Xbp1s up-regulates STUB1, which increases ubiquitination and degradation of FoxO1. This study identifies a novel pathway by which down-regulation of Xbp1s in HFpEF may result in up-regulation of FoxO1 and myocardial lipid accumulation in HFpEF. There are a number of issues that need to be addressed, as well as some missing experiments, that I believe are necessary to support the authors conclusions. These include:

- 1) To support the concept that Xbp1s is working through STUB1 to modify FoxO1 ubiquitination, it is important to know what happens to STUB1 in HFpEF? These experiments are missing.
- 2) Figure 1D shows that in XBP1s TG mice, a decrease in FoxO1 is seen. I believe it is important to show the data on what happens to FoxO1 protein in the XBP1s TG mice hearts subjected to HFpEF.
- 3) It is implied that the detrimental effects of FoxO1 in HFpEF are due to an increase in steatosis, and that activating Xbp1s is decreasing steatosis. However, the link between cardiac triglyceride levels and cardiac function is not clearly established in this study.
- 4) What does STUB1 overexpression do to lipid levels.
- 5) Figure 1B: Where are lipid levels in the non-HFpEF hearts?
- 6) Figure 1D: The effects of XBP1s TG on FoxO1 protein in HFpEF hearts are not shown.

7) Figure 3K and Supplemental Figure 1: In HFpEF hearts and cardiomyocytes an increase in CPT1B is reported. In addition, in FoxO1 ko hearts, FoxO1 decreased CPT1B. In these studies, CPT1B is portrayed as a lipogenic enzyme, when in fact it is a key enzyme involved in fatty acid oxidation. Normally, an increase in fatty acid oxidation would be expected to decrease steatosis. How do the authors explain this result? This is also relevant as it is suggested that fatty acid metabolism is down-regulated in the failing heart.

8) Figure 3H: It is proposed that the benefits of cardiomyocyte specific deletion of FoxO1 in HFpEF occurred in the absence of significant changes in cardiac hypertrophy. This is not obvious, and it appears that a decrease in cardiac hypertrophy may be evident.

9) The actual methodology used to produce HFpEF is not described in the paper, although it is clear that the authors are using the model they recently developed.

10) Figure 1F and G: These blots should be quantified.

11) Figure 2E: A marked increase in pdk4 was observed in LV HFpEF. What is the significance of this? This is a kinase that shuts down glucose oxidation. What is the significance of this, especially in the context of the authors suggesting that HFpEF results in the heart switching to glucose metabolism?

12) Figure 4F: This blot should be quantified.

13) In the introduction, the authors mention that Xbp1 can be activated upon ER stress. The authors also mention that ER stress is a common feature in disease-stressed cardiomyocytes such as HFpEF. Therefore, the concept that Xbp1 is low in HFpEF model and that the activation of Xbp1 can be cardioprotective is inconsistent with what the authors have suggested in the introduction. There is also no discussion to explain this dissociation between the previous literature and the findings of this study.

14) The role of FOXO1 in lipid homeostasis can be secondary to triggering the activity of pyruvate dehydrogenase kinase (PDHK), since PDHK inhibits cardiac glucose oxidation by inhibiting the activity of pyruvate dehydrogenase. As a result, cardiac fatty acid oxidation increases when FOXO1 is activated. Despite being a well-known mechanism for FOXO1 to influence cardiac energy metabolism, this mechanism is completely ignored and not explored in this study. More work is needed to investigate the involvement of this mechanism in the HFpEF model.

15) In a relevant point, the reduction in FOXO1 activity in the Xbp1s TG mice hearts could simply be explain by the fact that overexpression of Xbp1s induces a “pretend status of stress” in those Xbp1s TG mice hearts that leads to an overwhelming reliance on fatty acid. This increase in the reliance of fatty acid “overrides” role of FOXO1 in enhancing fatty acid oxidation, where FOXO1 is no longer needed and its activity is reduced. Therefore, this reduction in FOXO1 activity could completely be independent of Xbp1s.

16) The data with Oil Red O staining is not conclusive. These data do not really provide a clear insight into whether fatty acid oxidation is increased or decreased. These results could very well be interpreted as if there is an acceleration in fatty acid uptake and oxidation in cardiomyocytes which goes against the main conclusions of these study. More reliable approach should be employed to determine the impact of Xbp1s TG on fatty acid uptake and oxidation.

17) What is the ratio of BSA to oleic acid that was used for the AMVMs study?

Reviewer #3 (Remarks to the Author):

The manuscript by Schiattarella et al reports a mechanism of heart failure with preserved ejection fraction (HFpEF) involving stabilization of the transcription factor FoxO1 through suppression of Xbp1s and increased activity of its target Stub1. This work builds on a previous study by the same group demonstrating that suppression of Xbp1s is a critical driver of HFpEF pathogenesis. The current study implicates increased FoxO1 in HLPpEF pathology and also identifies the E3 ubiquitin ligase Stub1 as a direct target of Xbp1s involved in FoxO1 proteasomal degradation. Overall, the studies are convincing and the manuscript is well-written. The work has potential high impact in linking molecular regulation of cardiac metabolism by FoxO1 with pathology of HFpEF, for which there currently is no effective treatment.

Comments

1. Only male mice were used in the current study. A recent publication from the Hill group (Tong et al, reference 10) demonstrated that females were relatively protected in this HFpEF mouse model. Ideally data from females should be presented in the current work. At minimum, potential sex differences should be discussed.

2. Is FoxO1 protein increased in human HFpEF?

3. Much is known of FoxO1 target genes related to insulin signaling and metabolism. Additional information on the expression of some of these genes could be informative as to downstream mechanisms by which FoxO1 protein stabilization improves cardiac function, lipid accumulation and HFpEF pathology.

Response to Reviewers' Comments:**REVIEWER #1**

This manuscript describes a link between XBP1s and FOXO1 that is involved in regulating cardiac lipid accumulation and function in the context of the HFpEF heart failure model. This link is mediated through the posttranslational proteasomal degradation of FOXO1 through an XBP1s-dependent mechanism. The authors suggest that this degradation is mediated through XBP1s-dependent upregulation of the E3 ligase STUB1.

Overall, this manuscript covers the same XBP1s-dependent regulation of FOXO1 previously shown to regulate lipid metabolism in the liver (see Zhou et al (2011) Nat Med) in the context of cardiac lipid regulation in HFpEF. The XBP1s-FOXO1 link is not new, although its specific importance in the context of HFpEF has not been reported to my knowledge. That being said, there really isn't anything surprising in the link between XBP1s and FOXO1 in the context of lipid regulation in the heart based on published work from this group and many others.

We thank the reviewer for his/her insightful comments and questions and for recognizing the novelty of our investigation – in particular the XBP1s-STUB1 axis in the context of the devastating syndrome of HFpEF. To address all of the Reviewer's concerns, we have performed numerous additional experiments and analyses. In addition, we have modified the text extensively. We believe that the manuscript is significantly strengthened as a result. Below is a detailed response to each of the Reviewer's comments and observations.

Query 1. *The new information in this manuscript comes from the mechanism describing XBP1s-dependent suppression of FOXO1 through XBP1s-dependent upregulation STUB1. The data shown in Figure 4 does seem to suggest that STUB1 is a transcriptional target of XBP1s, although the authors should do the same experiments using other approaches that do not rely on XBP1s overexpression (e.g., treat with tunicamycin +/- an IRE1 inhibitor or Xbp1 siRNA).*

Response. We thank the reviewer for raising this important point. Following this suggestion, we have repeated the experiments identifying STUB1 as a direct transcriptional target of XBP1s in the absence of XBP1s over-expression by adenovirus. In particular, as shown in the **new Suppl. Figure 7**, NRVMs treated with the canonical ER stressor, tunicamycin, manifest a significant increase in STUB1 mRNA and protein levels. Importantly to confirm the specificity of XBP1s-dependency in STUB1 expression upon ER stress, we used two different approaches: NRVMs were treated with 1) MK-3946, a specific inhibitor of IRE1 α endoribonuclease activity (PMID: 22538852), the upstream activator of XBP1s or 2) siRNAs specific for XBP1s. Strikingly, with both approaches tunicamycin-dependent induction of STUB1 expression was drastically reduced. These data suggest that STUB1 expression is induced in the setting of ER stress in a XBP1s-dependent manner.

In addition, to further strengthen this novel finding, we evaluated STUB1 activity using a STUB1-luciferase assay upon tunicamycin treatment in the presence of XBP1s inhibition. As shown in the **new Suppl. Figure 7**, in both HEK cells and NRVMs, tunicamycin was able to increase STUB1-luciferase activity. Inhibition of XBP1s splicing with MK-3946 or XBP1s interference both resulted in significant reduction of STUB1-luciferase activity upon tunicamycin treatment. Collectively, these data, together with the previously presented data using specific XBP1s adenovirus, confirm that STUB1 is a novel, direct transcriptional target of XBP1s in cardiomyocytes.

Query 2. *However, the data in Figure 5 linking STUB1 to FOXO1 degradation is extremely weak. siRNA-dependent knockdown of Stub1 did not significantly change the degradation of FOXO1 induced by XBP1s overexpression in Figure 5B,C. I believe the authors are claiming that the fourth bar in Figure 5C is significantly different than the second bar in Figure C, but, even if they are statistically significant, this is not sufficient a difference to convince that STUB1 is required for XBP1s-dependent FOXO1 degradation. Moreover, the amount of XBP1s overexpression under these two conditions are not the same between these two conditions based on the gel shown in Figure 5B. Ultimately, the data included in this manuscript do not link STUB1 to the XBP1s-dependent degradation of FOXO1 significantly limiting the novelty of this work.*

Response. We apologize for the lack of clarity in the previous version of the manuscript and, at the same time, we thank the Reviewer for his/her comments suggesting a number of new experiments to further clarify the role of XBP1s-STUB1-FoxO1 axis in cardiomyocytes.

We agree that the representative blot in the previous version of Figure 5B was not clear. We have now replaced the old blot with a new one that is representative of the quantification reported. As shown in the **new Figure 5**, in the absence of STUB1, XBP1s-dependent FoxO1 degradation is significantly, albeit incompletely, reduced. We have attributed the lack of clarity in the previous version of the Western blot to an incomplete and inexact dosing in adenoXBP1s MOI, which we have addressed. Of note, we point out that in the same blot, it is evident that protein levels of STUB1 increased upon XBP1s overexpression. Confident in our results, we went on to demonstrate the STUB1 dependency of FoxO1 degradation upon XBP1s activation as follows:

1. We have confirmed and extended the role of STUB1 as an E3 ubiquitin ligase targeting FoxO1 in cardiomyocytes. As shown in the **new Figure 5**, NRVMs treated with cycloheximide in the absence of STUB1 manifested a significant decrease in the rate of FoxO1 protein turnover. The results of this classical experimental approach to evaluate protein turnover by blocking protein translation with cycloheximide lends additional support to a role for STUB1 in FoxO1 protein stability in cardiomyocytes.
2. Along the same lines, we have engineered an adenovirus coding for STUB1 (AdenoSTUB1). Infecting NRVMs with adenoSTUB1 resulted in a decrease in the steady-state levels of FoxO1 protein coupled with an increase in protein

ubiquitination compared with AdGFP-infected control cells (**new Suppl. Figure 8**), suggesting that STUB1 overexpression is functional and increases protein ubiquitination. Importantly, adenoSTUB1 infection did not result in reduction in FoxO1 mRNA level in NRVMs (**new Suppl. Figure 8**) suggesting that the STUB1-dependent reduction in FoxO1 protein level occurs via post-transcriptional mechanisms. Collectively, these results identify STUB1 as a critical element in FoxO1 protein stability in cardiomyocytes.

3. Pursuing the Reviewer's comments regarding the use of "other approaches that do not rely on XBP1s overexpression", we treated NRVMs with the ER stressor, tunicamycin, and evaluated FoxO1 protein levels. As shown in the **new Suppl. Figure 9**, tunicamycin treatment in NRVMs resulted in a reduction of FoxO1 protein levels. Interestingly, FoxO1 protein degradation upon tunicamycin treatment was partly restored by STUB1 knockdown (**new Suppl. Figure 9**). In addition, to confirm that tunicamycin-induced reductions in FoxO1 protein levels is dependent on XBP1s activation, we treated NRVMs with the IRE1 α inhibitor MK-3946. IRE1 α inhibition in cardiomyocytes significantly restored FoxO1 protein levels upon tunicamycin treatment. Importantly, all these changes in FoxO1 protein occurred in the absence of concomitant changes in its mRNA levels (**new Suppl. Figure 9**). In aggregate these data confirm that the activation of XBP1s in the setting of ER stress induces FoxO1 protein degradation in STUB1-dependent manner.
4. Finally, earlier findings from our group showed that HFD alone induced XBP1s (PMID: 30971818). To corroborate the notion that XBP1s induces STUB1, we detected a significant increase in STUB1 mRNA levels in HFD hearts relative to controls (**new Suppl. Figure 7**).

Considering that the main novelty of this paper (i.e., the XBP1s-STUB1 connection and its involvement in FOXO1 degradation) is not sufficiently demonstrated, I find that this manuscript is not ready for publication at this time. It is interesting that the XBP1s-FOXO1 regulatory axis contributes to cardiac function in the context of HFpEF, but in my opinion that link is not sufficient of an advance to merit publication in Nat Comm considering the significant published literature highlighting this same signaling mechanism in other tissues. I encourage the authors to better define the mechanism of the XBP1s-dependent FOXO1 degradation and if it does involve STUB1 to further pursue this relationship with additional experimental efforts that are more convincing than what is included in this manuscript. It is an interesting and important question that just isn't sufficient addressed in the included work.

We believe the results of all these experiments clarify the role of the XBP1s-STUB1 axis in FoxO1 degradation in cardiomyocytes. Despite this, we recognize, and have acknowledged in the text, that that this might not be the ONLY mechanism involved in FoxO1 degradation upon stress in cardiomyocytes; other mechanisms, which at this moment are beyond the scope of the current investigation, might play a role (please see page 14; lines 27-30)

REVIEWER #2

This study examined the potential mechanisms involved in the increase in FoxO1 and steatosis in a mouse model of HFpEF. A decrease in Xbp1s in HFpEF is proposed to decrease STUB1, a transcriptional target of Xbp1s. This resulted in a decreased ubiquitination of FoxO1 and an increase in nuclear FoxO1. Over-expression Xbp1s or knocking out FoxO1 resulted in an amelioration of the HFpEF phenotype. In addition, forced expression of Xbp1s in cardiomyocytes resulted in proteosomal degradation of FoxO1 due to activation of the E3 ubiquitin ligase STUB1. It is concluded that the Xbp1s-FoxO1 axis is a pivotal mechanism in the pathogenesis of HFpEF.

General Comments:

Previous studies by the authors have shown that in murine and human HFpEF a down-regulation of Xbp1s correlates with the development of the syndrome. This study adds to this previous observation by showing that Xbp1s up-regulates STUB1, which increases ubiquitination and degradation of FoxO1. This study identifies a novel pathway by which down-regulation of Xbp1s in HFpEF may result in up-regulation of FoxO1 and myocardial lipid accumulation in HFpEF. There are a number of issues that need to be addressed, as well as some missing experiments, that I believe are necessary to support the authors conclusions. These include:

We thank the Reviewer for his/her insightful comments and questions and for recognizing the novelty and importance of our investigation, stating that our “*study identifies a novel pathway by which down-regulation of Xbp1s in HFpEF may result in up-regulation of FoxO1 and myocardial lipid accumulation in HFpEF*”. To address all of the Reviewer’s comments, we have performed numerous additional experiments and analyses. In addition, we have modified the text and figures extensively. We believe that the manuscript is significantly strengthened as a result. Below is a detailed response to each of the Reviewer’s comments and observations.

Query 1. *To support the concept that Xbp1s is working through STUB1 to modify FoxO1 ubiquitination, it is important to know what happens to STUB1 in HFpEF? These experiments are missing.*

Response. We thank the Reviewer for his/her comment. We evaluated the mRNA level of STUB1 in HFpEF hearts. As shown in **Figure 4**, STUB1 was significantly down-regulated in HFpEF hearts compared to CTR hearts, similar to what we observed for XBP1s levels. These results are consistent with a model in which XBP1s down-regulation in HFpEF diminishes expression of its direct transcriptional target STUB1. In addition, we have added new data showing that HFD in isolation also increases STUB1 consistent with our previously published observation of an increase in XBP1s in HFD conditions (PMID: 30971818) (**new Suppl. Figure 7**).

Query 2. *Figure 1D shows that in XBP1s TG mice, a decrease in FoxO1 is seen. I*

believe it is important to show the data on what happens to FoxO1 protein in the XBP1s TG mice hearts subjected to HFpEF. / Figure 1D: The effects of XBP1s TG on FoxO1 protein in HFpEF hearts are not shown.

Response. We thank the Reviewer for raising this point. As suggested, we have included a new blot depicting FoxO1 levels in the hearts of all experimental groups. As shown in the **new Figure 1** FoxO1 protein levels were significantly reduced in XBP1 TG hearts subjected to CTR or HFpEF conditions.

Query 3. *It is implied that the detrimental effects of FoxO1 in HFpEF are due to an increase in steatosis, and that activating Xbp1s is decreasing steatosis. However, the link between cardiac triglyceride levels and cardiac function is not clearly established in this study.*

Response. [Please see also Response to Queries 15 and 16]. We value this observation. As the Reviewer suggested, cardiac dysfunction developed in obesity has been linked to the abnormal lipid metabolism/lipid accumulation referred to as lipotoxic cardiomyopathy (PMID: 19818871). A number of studies have shown that strategies aimed at reducing excess cardiac lipid can alleviate lipotoxic phenotypes and be beneficial for cardiac function (PMID: 17363697 - PMID: 22326951) suggesting that excessive fatty acid use is a culprit. The molecular mechanisms of lipotoxic cardiomyopathy are complex and still incompletely understood. Indeed, the specific link between cardiac lipid accumulation and the proposed cause of cardiomyopathy remains elusive. Respectfully, I submit that the answer to this important question goes beyond the scope of the present investigation. Here, we have investigated mechanisms whereby XBP1s exerts its cardioprotective effects in HFpEF. We have unveiled a novel XBP1s-STUB1-FoxO1 axis as a critical signaling pathway involved in the regulation of lipid accumulation/metabolism in HFpEF. We believe these data are important and will set the stage for more detailed investigation of alterations of cardiac metabolism in HFpEF, a devastating syndrome with no efficacious therapies. We have now discussed this issue in the amended version of the manuscript (please see page, 13; lines 26-30 and page, 15 lines 3-26).

Another point if I may. Increased cardiac fatty acid oxidation has long been considered to contribute to cardiac dysfunction in obese models. However, recent data from Rong Tian's group and others have suggested that increasing fatty acid oxidation by deletion of acetyl coenzyme A carboxylase 2 (ACC2) in the heart does not cause cardiac dysfunction in mice (PMID: 22730442 – PMID: 32597196). Taken together, these observations suggest that cardiac lipotoxicity is not attributable to increased fatty acid oxidation *per se*, but rather to an imbalance of fatty acid supply, storage, and use. Strikingly, this notion seems to hold true also in HFpEF. We have demonstrated that in cardiometabolic HFpEF (again, a completely unexplored area of investigation), cardiac lipid accumulates, fatty acid oxidation is reduced and these changes occur with the development of cardiac dysfunction (*Tong D, Schiattarella GG,....Hill JA, under peer review and new Suppl. Figure 6*). We hope that the Reviewer appreciates our effort in adding another piece to this complex puzzle.

Query 4. *What does STUB1 overexpression do to lipid levels?*

Response. We appreciate the Reviewer's comment regarding the role of STUB1 in regulating cardiomyocyte lipids. We agree with the Reviewer that addressing this question is important in the elucidation of STUB1 biology. However, respectfully, we also feel that addressing the role of STUB1 in lipid handling in cardiomyocytes would

stand as a full research project *per se* and goes beyond the scope of the present investigation focused on investigating the role of the XBP1s-FoxO1 axis in HFpEF. That said, to begin to address this interesting question, we engineered an adenovirus encoding STUB1 (AdSTUB1) in order to over-express STUB1 in cardiomyocytes. As shown in the panel on the left, STUB1 over-expression in NRVMs treated with the saturated lipid oleate significantly decreased lipid accumulation in

cardiomyocytes as evidenced by reduction in Oil Red O staining compared to AdGFP-infected control cells. Importantly, STUB1 over-expression was also associated with reduction in FoxO1 protein levels (**please see new Suppl. Figure 8**). These data suggest that STUB1 participates in lipid accumulation in NRVMs in accordance with previously published results demonstrating that STUB1 KO hearts contained extensive lipid accumulation (PMID: 23863712). We feel that these results, although interesting, are not central to the story presented in this manuscript. However, if the reviewer feels differently, we would be happy to incorporate them in the amended version of the manuscript.

Query 5. *Figure 1B: Where are lipid levels in the non-HFpEF hearts?*

Response. Given the very small amount of cardiac neutral lipid under baseline conditions, we omitted this information in the previous version of the manuscript. According to Reviewer's suggestion, we have now included cardiac TG levels in WT and XBP1s TG mice under CTR diet (**please see new Figure 1**). Importantly, XBP1s over-expression did not affect basal cardiac lipid levels under baseline conditions, suggesting that the presence of metabolic stress (i.e. HFpEF) is required to trigger the cardioprotective effects of XBP1s on lipid accumulation.

Query 6. *Figure 1D: The effects of XBP1s TG on FoxO1 protein in HFpEF hearts are*

not shown.

Response. Please refer to the Response to Query 2 above.

Query 7. *Figure 3K and Supplemental Figure 1: In HFpEF hearts and cardiomyocytes an increase in CPT1B is reported. In addition, in FoxO1 ko hearts, FoxO1 decreased CPT1B. In these studies, CPT1B is portrayed as a lipogenic enzyme, when in fact it is a key enzyme involved in fatty acid oxidation. Normally, an increase in fatty acid oxidation would be expected to decrease steatosis. How do the authors explain this result? This is also relevant as it is suggested that fatty acid metabolism is down-regulated in the failing heart.*

Response. [Please see also Response to Queries 3, 15 and 16]. We apologize for the confusion S/he is correct in that CPT1B is not a lipogenic enzyme. Figure 3K was mislabelled, we meant to present the lipogenic genes SREBP1 and FASN together; the FASN data were mislabeled as CPT1B. We greatly appreciate having this error pointed out!

To address the issues raised, we have now evaluated the mRNA of a panel of key enzymes involved in lipid oxidation (including CPT1b) in a larger number of HFpEF hearts. As shown in the **new Suppl. Figure 1** no statistically significant changes were observed compared with chow controls. This information, together with the new data directly showing that mitochondrial fatty acid oxidation is reduced in HFpEF hearts (**new Suppl. Figure 6**), confirm the complex cardiac energetic disturbances occurring in HFpEF

Query 8. *Figure 3H: It is proposed that the benefits of cardiomyocyte specific deletion of FoxO1 in HFpEF occurred in the absence of significant changes in cardiac hypertrophy. This is not obvious, and it appears that a decrease in cardiac hypertrophy may be evident.*

Response. We thank the Reviewer for bringing this up. To further corroborate the data indicating that, in the setting of HFpEF, cardiomyocyte-specific KO of FoxO1 does not impact LVH, we have evaluated the canonical LVH-related gene expression profile. As shown in the **new Suppl. Figure 5**, lack of FoxO1 does not affect the levels of ANP, BNP, β MHC and RCAN1.4 under CTR or HFpEF conditions. In other words, in HFpEF the same LVH gene expression signature was observed in WT and FoxO1 cKO hearts, suggesting that, at least upon the time of observation reported, genetic deficiency of FoxO1 does not significantly affect HFpEF-induced LVH in mice. Of note, we would like to emphasize that the role of FoxO1 in modulating LVH is highly dependent on the nature of the stress imposed on the ventricles. For example, we have recently demonstrated that “unlike T4 (thyroid hormone)-induced cardiomyocyte growth, knockdown of FoxO1 had little impact on the growth response elicited by other agonists” (PMID: 32439985) suggesting that in response to different stimuli, different mechanisms govern FoxO1’s role in LVH, an idea we find fascinating. We have discussed this point in the amended version of the manuscript (please see page, 15; lines 7-12).

Query 9. *The actual methodology used to produce HFpEF is not described in the paper, although it is clear that the authors are using the model they recently developed.*

Response. We apologize with the Reviewer for the missing information. We have now included the methodology used to induce HFpEF in mice in the amended version of the manuscript (please see page 17).

Query 10. *Figure 1F and G: These blots should be quantified.*

Response. Protein quantifications have been included in the amended version of the manuscript (**please see new Figure 1**)

Query 11. *Figure 2E: A marked increase in pdk4 was observed in LV HFpEF. What is the significance of this? This is a kinase that shuts down glucose oxidation. What is the significance of this, especially in the context of the authors suggesting that HFpEF results in the heart switching to glucose metabolism?*

Response. We thank the reviewer for raising this point. As correctly stated by the Reviewer, in HFpEF hearts we observed an increase in PDK4 levels. The increase in PDK4 level is a common feature of hearts subjected to metabolic stress (e.g. HFD). Given the fact that HFpEF is a model of “failure” that stems from metabolic stress, it is not particularly surprising that PDK4 levels increase in HFpEF hearts. We have now confirmed this result also at the protein level and have incorporated cardiomyocyte-specific PDK4 transgenic (PDK4 TG) hearts as positive control for the PDK4 protein band (**please see new Suppl. Figure 6**).

Regarding the role of mitochondrial glucose oxidation and fatty acid oxidation in HFpEF hearts, we would like to communicate to this Reviewer that we have a manuscript currently under review that specifically address this issue (*Tong D, Schiattarella GG, ...Hill JA, under review*). Specifically, we observed that both glucose oxidation and FAO are impaired in HFpEF hearts. If the Reviewer feels appropriate, we will be willing to confidentially send this manuscript to him/her.

Coming back to the present investigation, the primary focus of which was to investigate the cardioprotective effects of the XBP1s-FoxO1 axis in HFpEF, we have now included in the amended version of the manuscript new data demonstrating that in HFpEF mitochondria both pyruvate-driven respiration and fatty acid-driven respiration are substantially impaired (**new Suppl. Figure 6**). Interestingly, in FoxO1 cKO mice subjected to HFpEF, we observed a trend toward amelioration of mitochondrial fatty acid oxidation in the absence of changes in pyruvate oxidation (**please see new Suppl. Figure 6**) or PDK4 levels (**please see new Suppl. Figure 6**). Collectively, these results imply that 1) a major disruption in energetic substrates utilization occurs in HFpEF hearts, and 2) lack of FoxO1 tends to improve this energetic disturbance acting through the restoration of fatty acid oxidation reducing, concomitantly, HFpEF-induced

lipotoxicity. We now comment on this issue in the amended version of the manuscript (please see page, 13; lines 26-30 and page, 15 lines 3-26).

Query 12. *Figure 4F: This blot should be quantified.*

Response. Protein quantification has been included in the amended version of the manuscript (**please see new Figure 4**).

Query 13. *In the introduction, the authors mention that Xbp1 can be activated upon ER stress. The authors also mention that ER stress is a common feature in disease-stressed cardiomyocytes such as HFpEF. Therefore, the concept that Xbp1 is low in HFpEF model and that the activation of Xbp1 can be cardioprotective is inconsistent with what the authors have suggested in the introduction. There is also no discussion to explain this dissociation between the previous literature and the findings of this study.*

Response. We apologize that we have not made clear this point. Activation of the XBP1s arm of the UPR is a major response in cardiomyocytes to cope with stress of various origins. Indeed, XBP1s activation has been observed in virtually all cardiovascular diseases (e.g. pressure overload-induced cardiac hypertrophy and myocardial infarction). Recently, in a series of manuscripts (Schiattarella GG,..... Hill JA, Nature 2019, Tong D*, Schiattarella GG*.... Hill JA, Circulation 2019) we reported what has never been observed before in the cardiovascular field: inflammation-dependent suppression of the XBP1s pathway in both experimental and human HFpEF. This was a very surprising finding at the time and prompted us to investigate the underlying mechanisms. We went on to demonstrate that restoring XBP1s in HFpEF cardiomyocytes greatly ameliorates the HFpEF syndrome. In the present investigation, we set out to delineate downstream mechanisms underlying the XBP1s cardioprotective effects. Importantly, the notion that XBP1s activation confers cardiac benefits holds true also for cardiovascular diseases other than HFpEF, such as ischemic cardiomyopathy (PMID: 24630721). We hope we have clarified this point in our amended text.

Query 14. *The role of FOXO1 in lipid homeostasis can be secondary to triggering the activity of pyruvate dehydrogenase kinase (PDHK), since PDHK inhibits cardiac glucose oxidation by inhibiting the activity of pyruvate dehydrogenase. As a result, cardiac fatty acid oxidation increases when FOXO1 is activated. Despite being a well-known mechanism for FOXO1 to influence cardiac energy metabolism, this mechanism is completely ignored and not explored in this study. More work is needed to investigate the involvement of this mechanism in the HFpEF model.*

Response. [Please see also Response to Query 11]. We thank the reviewer for raising this important point. We agree with the Reviewer that the investigation of cardiac metabolic alterations in HFpEF (i.e. glucose oxidation and fatty acid oxidation) will provide fundamental insights in the understanding of HFpEF pathogenesis. To this end, as mentioned above in the response to your Query 11, we have conducted an in-depth

evaluation of cardiac mitochondrial structural and functional abnormalities in HFpEF. This manuscript is currently under peer revision (*Tong D, Schiattarella GG,....Hill JA*). As you correctly point out, under certain conditions FoxO1 activation results in increased activity of PDK4 which suppresses glucose oxidation. However, in our model of cardiometabolic HFpEF, genetic deletion of FoxO1 has no impact on the upregulation of PDK4 (**new Suppl. Figure 6**). As a result, mitochondrial pyruvate oxidation is reduced in HFpEF with or without FoxO1 suggesting that the effects on FAO we observe are not secondary to PDH inhibition (**new Suppl. Figure 6**). Interestingly, we also observe a reduction in fatty acid oxidation under HFpEF conditions (**new Suppl. Figure 6**), which might contribute to the cardiac steatosis and lipotoxicity we observe. In addition, we were able to show that genetic silencing of FoxO1 in HFpEF cardiomyocytes partially restores fatty acid oxidation (**new Suppl. Figure 6**) diminishing HFpEF cardiac steatosis (**Figure 3**). In aggregate, these data point to unique alterations in cardiac metabolism of HFpEF hearts that are ameliorated by FoxO1 inactivation.

Query 15. *In a relevant point, the reduction in FOXO1 activity in the Xbp1s TG mice hearts could simply be explain by the fact that overexpression of Xbp1s induces a “pretend status of stress” in those Xbp1s TG mice hearts that leads to an overwhelming reliance on fatty acid. This increase in the reliance of fatty acid “overrides” role of FOXO1 in enhancing fatty acid oxidation, where FOXO1 is no longer needed and its activity is reduced. Therefore, this reduction in FOXO1 activity could completely be independent of Xbp1s.*

Response. Your point is well taken. We would like to point out that we have provided a number of pieces of evidence, both *in vitro* and *in vivo*, supporting the conclusion that, in cardiomyocytes, over-activation of XBP1s leads to FoxO1 degradation and consequently to its reduction in activity. Please note that in our model, the reduction in FoxO1 activity is directly related to reduction in FoxO1 protein levels and does not exclusively occur by “suboptimal activation” of FoxO1. Therefore, respectfully, our data strongly suggest that the observed reduction in FoxO1 activity is greatly dependent on XBP1s.

In response to your comment regarding the possibility that XBP1s overexpression might induce a stress response in cardiomyocytes that will increase fatty acid oxidation, we have evaluated fatty acid-driven respiration in cardiac mitochondria from XBP1s TG mice. Similar to what has been observed in NRVMs-over-expressing XBP1s (**new Suppl. Figure 2**), mitochondrial fatty acid oxidation is not increased in XBP1s TG hearts (**new Suppl. Figure 7**). These data suggest that XBP1s over-expression, *per se*, does not induce “cardiomyocyte stress” and does not alter cardiomyocyte energetic substrate utilization. In addition, the fact that cardiomyocyte-specific over-expression of XBP1s does not elicit major detrimental structural or functional consequences in the heart is also supported by the fact that, in the time frame analyzed, XBP1s hearts do not exhibit alterations in LV systolic and diastolic function (**new Suppl. Figure 2**).

Query 16. *The data with Oil Red O staining is not conclusive. These data do not really provide a clear insight into whether fatty acid oxidation is increased or decreased.*

These results could very well be interpreted as if there is an acceleration in fatty acid uptake and oxidation in cardiomyocytes which goes against the main conclusions of these study. More reliable approach should be employed to determine the impact of Xbp1s TG on fatty acid uptake and oxidation.

Response. [Please see also Response to Query 15]. We agree with the reviewer that Oil Red O staining provides an immediate qualitative assessment of cardiac steatosis, but, if used alone, is not sufficient to infer the “lipid” status of heart. For this reason, we have paired this staining with the quantification of neutral lipids in the heart (**Figure 1**). In addition, following the Reviewer’s suggestion, we have evaluated in XBP1s TG hearts the mRNA levels of the key genes involved in lipid storage/transport (**new Suppl. Figure 2**). The majority of these were significantly increased in WT HFpEF hearts. Consistent with our model, XBP1s over-expression induced a significant reduction in this gene set (**new Suppl. Figure 2**), suggesting that in XBP1s TG hearts under HFpEF conditions lipid accumulation is reduced. In aggregate, these data allowed us to conclude that in XBP1s TG mice subjected to HFpEF a reduction in cardiac steatosis is observed. Regarding the mechanisms whereby XBP1s promotes lipid clearance, we have provided evidence that XBP1s overexpression, per se, does not increase fatty acid oxidation in both NRVMs and cardiac mitochondria isolated from XBP1s TG mice. These data, coupled with the increase in FoxO1 degradation, prompted us to conclude that XBP1s over-expression ameliorated cardiac lipotoxicity by suppression of FoxO1.

Query 17. *What is the ratio of BSA to oleic acid that was used for the AMVMs study?*

Response. As stated in the Methods section, NRVMs were treated with BSA/Oleic acid complexes solution purchased from Sigma-Aldrich (Sigma-Aldrich, O3008-5ML). According to the manufacturer’s specification, the composition is 2 mol of oleic acid per 1 mol of BSA. We have now included this information in the amended version of the manuscript (please see page 18).

REVIEWER #3

The manuscript by Schiattarella et al reports a mechanism of heart failure with preserved ejection fraction (HFpEF) involving stabilization of the transcription factor FoxO1 through suppression of Xbp1s and increased activity of its target Stub1. This work builds on a previous study by the same group demonstrating that suppression of Xbp1s is a critical driver of HFpEF pathogenesis. The current study implicates increased FoxO1 in HLPfEF pathology and also identifies the E3 ubiquitin ligase Stub1 as a direct target of Xbp1s involved in FoxO1 proteasomal degradation. Overall, the studies are convincing, and the manuscript is well-written. The work has potential high impact in linking molecular regulation of cardiac metabolism by FoxO1 with pathology of HFpEF, for which there currently is no effective treatment.

We thank the Reviewer for his/her insightful comments and questions and for recognizing the quality of our data and the novelty and importance of our investigation stating that “*the studies are convincing, and the manuscript is well-written. The work has potential high impact in linking molecular regulation of cardiac metabolism by FoxO1 with pathology of HFpEF*”. To address all of the reviewer’s comments, we have performed additional experiments, and we have modified the text according to the reviewer’s suggestions. We believe that the manuscript is significantly improved as a result. Below is a detailed response to each of the reviewer’s comments and observations.

Query 1. *Only male mice were used in the current study. A recent publication from the Hill group (Tong et al, reference 10) demonstrated that females were relatively protected in this HFpEF mouse model. Ideally data from females should be presented in the current work. At minimum, potential sex differences should be discussed.*

Response. We thank the reviewer for raising this important point. As correctly pointed out, in our recent work we have demonstrated that female mice are relatively protected from experimental HFpEF. In particular, we have shown that this protection is dependent by the lower inflammatory burden in females which, as consequence, results in less XBP1s suppression. Therefore, in the present investigation, in order to study the mechanisms by which XBP1s restoration ameliorates the HFpEF phenotype in mice, we only studied male mice. This was by design. However, we certainly recognize that sex-differences in HFpEF pathophysiology are an underdeveloped area of investigation and that elucidation of sex-neutral signaling pathways involved in HFpEF pathogenesis might help in identifying novel therapeutic targets for this prevalent and devastating syndrome. We now comment on this issue in the amended version of the manuscript (please see page 16; lines 17-23).

Query 2. *Is FoxO1 protein increased in human HFpEF?*

Response. We certainly agree that the evaluation of a newly discovered signaling pathway in experimental models of cardiovascular disease would be strengthened by confirmatory results in human cardiac samples. Unfortunately, and different from HFrEF for which usually ample tissue specimens are available, the availability of HFpEF human cardiac tissue is very limited in amount given the lack of clearly recognized indications for endomyocardial biopsy in this condition. Cardiac sampling retrieves only a few milligrams of myocardial tissue. Therefore, only limited RNA or protein analysis can be accomplished with these samples. Indeed, in our recent publication (Schiattarella GG, ... Hill JA, Nature 2019), we were able to show that the very same signaling pathway we unveiled was dysregulated in both experimental and clinical HFpEF using human HFpEF endomyocardial biopsies. These data provided corroborating evidence for the veracity of our model. Based on these considerations and due to the severe impact of COVID-19 on our lab and clinic, this time, unfortunately, we were unable to evaluate the level of FoxO1 in human samples. That said, we point you to a recent publication from David Kass's lab (Circulation 2020, in press). In this manuscript, transcriptomic analysis of human cardiac tissue from a large, obese (i.e. similar to our experimental model) HFpEF cohort of patients identified fatty acid dysregulation and alterations in ER/UPR markers as two of the top differentially regulated mechanisms in HFpEF, supporting in human samples our experimental evidence and conclusion. We now comment on this issue in the amended version of the manuscript (please see page 14).

Query 2. *Much is known of FoxO1 target genes related to insulin signaling and metabolism. Additional information on the expression of some of these genes could be informative as to downstream mechanisms by which FoxO1 protein stabilization improves cardiac function, lipid accumulation and HFpEF pathology.*

Response. We thank the reviewer for raising this point. As correctly stated, we (PMID: 22326951) and others have shown that downstream target genes of FoxO1 regulate both glucose and lipid homeostasis in the heart, substantially impacting cardiac energetic homeostasis. In the present investigation, we have now extended our understanding of FoxO1 biology in cardiac metabolism into the setting of HFpEF. We have now included in the amended version of the manuscript, new data demonstrating that, in HFpEF mitochondria, both pyruvate-driven respiration and fatty acid-driven respiration are substantially impaired (**please see new Suppl. Figure 6**). Interestingly, in FoxO1 cKO mice subjected to HFpEF we observed a trend toward amelioration of mitochondrial fatty acid oxidation in the absence of changes in pyruvate oxidation (**please see new Suppl. Figure 6**) or PDK4 level (**please see new Suppl. Figure 6**). Collectively these results reveal that 1) a major disruption in energetic substrate utilization occurs in HFpEF hearts, and 2) lack of FoxO1 tends to improve this energetic disturbance acting through restoration of fatty acid oxidation with consequent decreases in HFpEF-induced lipotoxicity. We now comment on this issue in the amended version of the manuscript (please see page, 13; lines 26-30 and page, 15 lines 3-26).

REVIEWERS' COMMENTS

Reviewer #1 (Remarks to the Author):

In the revised manuscript, the authors performed a number of additional experiments to try and strengthen the connection between STUB1 and posttranslational regulation of FOXO1. Notably, the authors show that Stub1 is induced by ER stress through a mechanism sensitive to an IRE1 RNase inhibitor and that this genetic manipulation of Stub1 (overexpression or knockdown) alters FOXO1 protein levels in cellular models. With these experiments the authors have done enough to convince that STUB1 has some role in the posttranslational regulation of FOXO1, although it is clearly not the only mechanism responsible for this regulation. As such, the major issues with the manuscript brought up in my initial review have been addressed.

Reviewer #2 (Remarks to the Author):

None

Reviewer #3 (Remarks to the Author):

My comments have been addressed and the MS is improved. I have no additional concerns.

Response to Reviewers' Comments:**REVIEWER #1**

In the revised manuscript, the authors performed a number of additional experiments to try and strengthen the connection between STUB1 and posttranslational regulation of FOXO1. Notably, the authors show that Stub1 is induced by ER stress through a mechanism sensitive to an IRE1 RNase inhibitor and that this genetic manipulation of Stub1 (overexpression or knockdown) alters FOXO1 protein levels in cellular models. With these experiments the authors have done enough to convince that STUB1 has some role in the posttranslational regulation of FOXO1, although it is clearly not the only mechanism responsible for this regulation. As such, the major issues with the manuscript brought up in my initial review have been addressed.

Response. We thank the Reviewer for his/her comments and suggestions that have contributed to improving our manuscript, and we are pleased that he/she is satisfied now.

REVIEWER #2

None.

Response. We thank the Reviewer for his/her comments and suggestions that have contributed to improving our manuscript, and we are pleased that he/she is satisfied now.

REVIEWER #3

My comments have been addressed and the MS is improved. I have no additional concerns.

Response. We thank the Reviewer for his/her comments and suggestions that have contributed to improving our manuscript, and we are pleased that he/she is satisfied now.